

# Capacity for heat absorption by the wings of the butterfly *Tirumala limniace* (Cramer)

Huaijian Liao[1,2], Ting Du[1], Yuqi Zhang[3], Lei Shi[1], Xiyu Huai[1], Chengli Zhou[1] and Jiang Deng[1]

[1] Research Institute of Resources Insects, Chinese Academy of Forestry, Kunming, Yunnan, People's Republic of China
[2] Institute of Leisure Agriculture, Jiangsu Academy of Agricultural Sciences, Nanjing, Jiangsu, People's Republic of China
[3] College of Life Science, Southwest Forestry University, Kunming, Yunnan, People's Republic of China

Corresponding authors
Huaijian Liao,
huaixiyu_08@126.com
Lei Shi, leishi@139.com

## ABSTRACT

Butterflies can directly absorb heat from the sun via their wings to facilitate autonomous flight. However, how is the heat absorbed by the butterfly from sunlight stored and transmitted in the wing? The answer to this scientific question remains unclear. The butterfly *Tirumala limniace* (Cramer) is a typical heat absorption insect, and its wing surface color is only composed of light and dark colors. Thus, in this study, we measured a number of wing traits relevant for heat absorption including the thoracic temperature at different light intensities and wing opening angles, the thoracic temperature of butterflies with only one right fore wing or one right hind wing; In addition, the spectral reflectance of the wing surfaces, the thoracic temperature of butterflies with the scales removed or present in light or dark areas, and the real-time changes in heat absorption by the wing surfaces with temperature were also measured. We found that high intensity light (600–60,000 lx) allowed the butterflies to absorb more heat and 60–90° was the optimal angle for heat absorption. The heat absorption capacity was stronger in the fore wings than the hind wings. Dark areas on the wing surfaces were heat absorption areas. The dark areas in the lower region of the fore wing surface and the inside region of the hind wing surface were heat storage areas. Heat was transferred from the heat storage areas to the wing base through the veins near the heat storage areas of the fore and hind wings.

## INTRODUCTION

Sunlight is the most important source of energy and it supports the survival and reproduction of most creatures on Earth. Butterflies can directly utilize the heat from the sun to facilitate their autonomous flight. Previous studies have shown that butterflies must directly absorb the heat from sunlight to increase their thoracic temperature above that of their surroundings in order to allow autonomous flight to occur (*Kemp & Krockenberger, 2004*; *Barton, Porter & Kearney, 2014*; *Bonebrake et al., 2014*;

*Kleckova, Konvicka & Klecka, 2014*; *Mattila, 2015*; *Liao et al., 2017*). Autonomous flight is required for the reproductive activities of butterflies, including dispersal, courtship, mating, and oviposition behaviors (*Shreeve, 1992*; *Bennett, Smith & Betts, 2012*). Therefore, heat absorption is an indispensable prerequisite for the reproductive success of butterflies (*Gibbs, Van Dyck & Karlsson, 2010*; *Westerman, Drucker & Monteiro, 2014*; *Gibbs, Van Dyck & Breuker, 2018*).

Butterflies primarily absorb heat via their wings (*Shanks et al., 2015*; *Han et al., 2016*; *Niu et al., 2016*). When adult butterflies require heat for autonomous flight, they adjust their body posture and either fully expand their wings or angle them to allow the optimal absorption of sunlight, where the heat absorbed is then transferred to the thoracic muscles to enable flight (*Heinrich, 1990*; *Huey, Hertz & Sinervo, 2003*). Some studies indicate that the angle of the incoming sunlight and the wing opening angle can both significantly influence the rate and amount of heat absorption in butterflies (*Plattner, 2004*; *Shanks et al., 2015*; *Niu et al., 2016*). Indeed, heat absorption is important, but it has to be modulated via behavior to avoid overheating (*Kingsolver, 1985*).

Based on a study of a new mechanism of behavioral thermoregulation, *Kingsolver (1985)* suggested that *Pieris* butterflies use their predominantly white wings as solar reflectors to reflect solar radiation onto the body and to increase their body temperature. Some studies also indicate that butterflies rely on "photonic crystal" structures in their wings to absorb heat (*Li et al., 2004*; *Han et al., 2013*, *2015a*; *Luohong, 2014*). Therefore, the transfer of heat in a butterfly not only depends on reflection and absorption from the wing surface, but it also is due to the internal tissue in the wing. Thus, what is the specific tissue involved with heat transfer in the wing? The answer to this question remains unclear. Previous studies of solar heat absorption by butterfly wings have focused on the forewings (*Devries, 2015*; *Wu et al., 2015*) whereas none have considered the hind wings. As a consequence, it is not clear whether the fore wings or hind wings have a greater capacity for heat absorption.

Light intensity also significantly affects heat absorption by butterflies. In the butterfly *Heteronympha merope*, as the intensity of solar radiation increases, the thoracic temperature excess increases significantly regardless of wings being fully extended or closed (*Barton, Porter & Kearney, 2014*). *Kingsolver (1983)* found that short-term meteorological variations, such as intermittent cloudy periods, can rapidly reduce the body temperature excess and flight activity to zero in *Colias* butterflies. The light intensity also significantly influences the ability of the butterfly *Tirumala limniace* to absorb heat, where it was shown that the equilibrium thoracic temperature of adults was higher when exposed to 2,240 lx compared with other light intensities and the time required to reach it was shorter. In addition, the thoracic temperature excess and rate of thoracic temperature increase were higher and achieved more quickly (*Liao et al., 2017*), the sunlight intensity is about 2,240 lx from 7.00 am–8.00 am in summer in Yuanjiang county, Yunnan province, China.

Previous studies have shown that color significantly affects the absorption of heat in butterflies (*Schmitz, 1994*; *Van Dyck, Matthysen & Dhondt, 1997*; *Brashears, Aiello & Seymoure, 2016*; *Stuart-Fox, Newton & Clusella-Trullas, 2017*). Some studies indicate

that more heat is absorbed when the wings of butterflies are darker, (*Kingsolver, 1987*; *Schmitz, 1994*; *Berwaerts et al., 2001*; *Clusella Trullas, Van Wyk & Spotila, 2007*). Some butterflies exhibit wing surface melanism to increase their ability to absorb heat (*Guppy, 1986*; *Clusella Trullas, Van Wyk & Spotila, 2007*), but wing melanization is not directly functional to increasing heat absorption, as melanin is involved in numerous functions (*Stuart-Fox, Newton & Clusella-Trullas, 2017*). Compared with the distal wing area, melanism on the wing surface near the body is more conducive to heat absorption by butterflies (*Kingsolver, 1987*; *Brashears, Aiello & Seymoure, 2016*). However, few studies have considered the specific wing surface areas that are used for heat absorption.

In this study, we aimed to understand the effects of the light intensity and wing opening angle on heat absorption, as well as clarifying the differences in the heat absorption capacity of the fore and hind wings, the distribution of heat absorption and non-heat absorption areas on the wing surfaces, and whether the wing has the ability to store heat temporarily and the location of the heat storage areas on the wing surfaces in order to identify the wing tissue that serves as the heat transfer channel. Elucidating the heat absorption capacity of butterflies may provide a deeper understanding of the utilization of solar heat by insects.

*Tirumala limniace* (Cramer) is a typical butterfly that directly utilizes solar heat. In the field, we have investigated that adults show high flight activity on the sunny days, but on cloudy and rainy days, the butterflies do not fly even when the air temperature is high. The wings of the male and female butterflies have the same morphology and the wing surface only comprises black (black background) and light blue (light blue bands) areas (*Chen et al., 2008*). Hence, according to the color depth, the wing surface can be divided into dark and light areas, while the wings of other butterflies in Yuanjiang County have more colors composition. Thus, the color composition of the wing surface is suitable for studying the heat absorption capacity of butterflies. At 243–2,240 lx, high light intensity enabled adult butterflies to quickly absorb more heat than at low light intensities to elevate their thoracic temperature to above that of the environmental temperature and to take off earlier than at low light intensities (*Liao et al., 2017*). However, the light intensity range considered in a previous study by *Liao et al. (2017)* was too narrow to determine the appropriate light intensity for heat absorption by the adult *T. limniace* butterfly. In China, *T. limniace* is mainly distributed in the dry hot valleys of south Yunnan Province, Guangxi Province, and Guangdong Province (*Chen et al., 2008*). Yuanjiang County, Yunnan Province, China, which was the site of the artificial breeding ground, is also in the actual habitat range for this butterfly. In this study, according to the natural light intensity in Yuanjiang County, a light intensity range of 600–60,000 lx was selected to study the heat absorption capacity of *T. limniace* butterflies in order to investigate the mechanism responsible for solar heat utilization by insects.

Thus, in the present study, we measured the absorption of heat by the butterfly *T. limniace* at different light intensities and wing opening angles in order to determine the most appropriate light intensity and wing opening angle for heat absorption. The thoracic temperature was measured in a butterfly with only one right fore wing or one right hind wing to elucidate the difference in the heat absorption capacity of the fore and hind

wings. The spectral reflectance was measured for the wings to identify the distribution of the heat absorption and non-heat absorption areas on the fore and hind wings. In order to demonstrate that dark areas are responsible for heat absorption on butterfly wings, we investigated heat absorption in butterflies with the scales removed or present in light or dark areas, respectively. To identify the wing surface areas involved with heat storage and transfer after heat absorption by butterfly wings, we monitored the real-time temperature changes on the wing surface during heat absorption in adult *T. limniace*. Once we identified the heat storage areas and non-storage areas, we also monitored real-time temperature changes in these areas, as well as the wing veins involved with heat transfer, to further determine the heat storage areas and the transfer channel on the wing surface. Based on the results, we obtained a clear understanding of the heat absorption capacity of the wings in the butterfly *T. limniace* to facilitate further research into the mechanism responsible for the utilization of solar heat by insects.

## MATERIALS AND METHODS

### Insects

*Tirumala limniace* pupae were purchased from the Ornamental Insect Technology Development, Kunming Zhonglin Co. Ltd., Kunming, China. The pupae were placed on a cylindrical net (height: 65 cm, diameter: 50 cm) on top of a towel so they could readily spread their wings after emergence. Subsequently, the pupae were transferred to a room at 25 ± 1 °C with a relative humidity of 75% ± 5% under a 14:10 light:dark photoperiod. After the adults emerged, the females and males were reared separately in two different cages to prevent mating. A total of five-day old unmated adults (306 female and 308 male adults) were used in this study. All the butterflies used in the experiment were living.

In China, *T. limniace* is mainly distributed in the dry hot valleys of south Yunnan Province, Guangxi Province, and Guangdong Province. Yuanjiang County, Yunnan Province, China, which was the artificial breeding ground, is also in their normal range habitat. This butterfly is important for ecological and environmental quality monitoring in south China. The optimal temperature range for survival and flight is 25–36 °C. During the first and second day after emergence, adult *T. limniace* gradually unfurl their wings and exhibit low flight activity. Their flight activity increases significantly from 3 days of age and the peak flight activity occurs from 4 to 7 days of age during the summer and autumn when the thermal absorption activity is also very high. Thus, five-day aged unmated adults were used in this study.

### Effects of light intensity and wing opening angle on the heat absorption capacity

We investigated the effects of the light intensity and wing opening angle on the heat absorption capacity of adult *T. limniace* butterflies. Yuanjiang County, Yunnan Province, China, is a suitable habitat for the butterfly *T. limniace*, and its population is mainly observed from July to October every year. Thus, the natural light intensity on September 11, 2017 in Yuanjiang County, Yunan Province, China, was selected as the reference

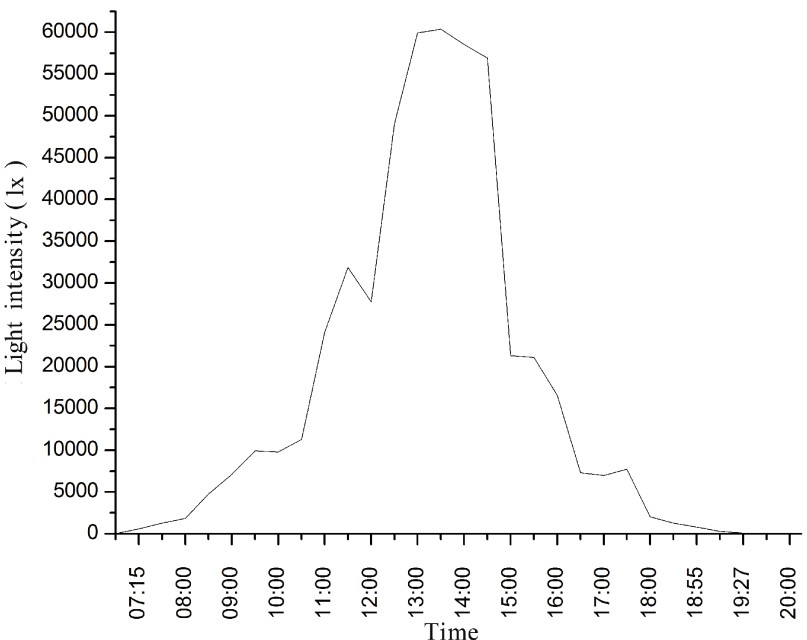

**Figure 1 Natural light intensity on September 11, 2017 in Yuanjiang County, Yunnan Province, China.**                                                

standard for the light intensity in this study because the skies were clear and cloudless on this day. Based on the natural light intensity levels between 7.00 and 20.00 h on September 11, 2017 (Fig. 1), six light intensity levels were selected for this study, that is, 600, 5,000, 15,000, 25,000, 40,000, and 45,000 lx. The left and right wings form a specific angle when they are open which angle ranges from 0° to 180°. Based on this angle range, 10 wing opening angles were tested in this study, that is, 15°, 30°, 45°, 60°, 75°, 80°, 90°, 120°, 150°, and 180°. An orthogonal design was used in this experiment to study the effects of the light intensity and wing opening angle on heat absorption by the butterfly wings.

According to the specific angle required, a plane with the selected angle was cut with a knife along the middle line of a white plastic foam board (side length: 10 cm, thickness: six cm) and the adult *T. limniace* wings were fixed on this plane. Thus, a new unmated five-day-old adult butterfly was fixed on the plane at a specific wing opening angle using double-sided adhesive tape. The plane with the affixed adult butterfly was placed in a dark room (length: 100 cm, width: 80 cm, height: 80 cm). The walls of the dark room were made of black-out cloth and the front wall of the dark room comprised a curtain made of awning cloth to facilitate the placement of butterflies. A 75 W halogen lamp (OSRAM-64841FLSP; Osram China Lighting Ltd., Shanghai, China) was hung 60 cm above the bottom of the dark room. The light intensity of the lamp was adjusted with a potentiometer (CHILFU; Shanghai Wenfu Electric Co. Ltd., Shanghai, China).

Before turning the lamp on, the butterfly fixed on the plane was maintained in the dark room for 15 min in order to stabilize its body temperature to avoid the influence of body temperature on heat absorption by the butterfly. After switching the lamp on, the

thoracic temperature was recorded every 10 s until the temperature was maintained at a set temperature for 90 s, thereby indicating that the thoracic temperature of the butterfly had reached equilibrium under the specific light intensity. The thoracic temperature was measured using a thermocouple thermometer (Thermometer GM1312; Shenzhen Jumaoyuan Science and Technology Co. Ltd., Shenzhen, China). A thermistor was inserted through the mesoscutellum and placed in or near the dorsal longitudinal muscles to record the thoracic temperature. During each run of the experiment, we recorded the equilibrated thoracic temperature, rate of thoracic temperature increase (total change in temperature over time), $\Delta T$ as the difference in temperature between the thorax and the ambient temperature, and the time when the thoracic temperature reached equilibrium. Three female and three male adult butterflies were tested for each orthogonal treatment with the different light intensities and wing opening angles. In total, 180 female and 180 male butterflies were tested in this experiment.

## Difference in the heat absorption capacity of the fore and hind wings

To determine the difference in the heat absorption capacity of the fore and hind wings, we measure the thoracic temperature in a butterfly with only one right fore wing or only one right hind wing under six light intensities, that is, 600, 5,000, 15,000, 25,000, 40,000 and 45,000 lx. The two hind wings and left fore wing were cut from a new unmated five-day old adult butterfly using surgical scissors so only the right fore wing remained (fore wing treatment). The two fore wings and left hind wing were cut from another new unmated 5-day-old adult butterfly using surgical scissors so only the right hind wing remained (hind wing treatment). The butterfly was then fixed on a horizontal plane (side length: 10 cm, thickness: six cm) without any angle, before it was placed and maintained in the dark room for 15 min so its body temperature could stabilize before turning the lamp on. After switching the lamp on, during each run of the experiment, we recorded the equilibrated thoracic temperature, the rate (°C/min) of thoracic temperature increase, $\Delta T$, and the time required to reach equilibrium. In this experiment, we tested six female and six male butterflies with only one right fore wing at each light intensity, and six female and six male butterflies with only one right hind wing at each light intensity. In total, we tested 36 female and 36 male butterflies in the fore wing treatment, and 36 female and 36 male butterflies in the hind wing treatment. We also measured the wing expanse for both the fore and hind wings of the adult butterflies using 37 female and 34 male adult butterflies.

## Reflectance of light and dark areas

To determine the distribution of the heat absorption and non-heat absorption areas on the fore and hind wings, we measured the spectral reflectance from the wings of adult butterflies. A higher spectral reflectance indicates weaker light absorption and heat absorption, whereas lower spectral reflectance denotes stronger light absorption and heat absorption. The wing of the *T. limniace* butterfly appears the same in both sexes, but they differ because there is a scaly bag with a protruding ear in the middle of the $Cu_2$ chamber in the male butterfly. The wing surface is black with many transparent azure

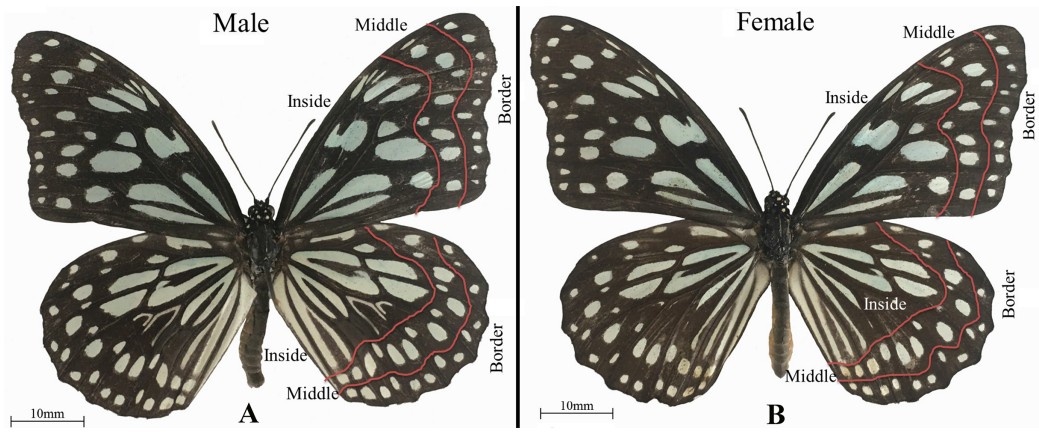

**Figure 2 The wing surface was divided into border, middle, and inside parts to obtain spectral reflectance measurements from the wings of adult *Tirumala limniace* butterflies.** (A) Male, (B) female.

markings (Fig. 2). Therefore, the wing surface can be divided into light areas and dark areas according to the color shade on the wing surface.

The right fore and right hind wings were cut from a female butterfly and a male butterfly. During the cutting process, the integrity of the wings was maintained as far as possible and we prevented scale removal from the wings. The wings were then placed in the dark room. The light intensity in the dark room was set to 5,000 lx. The spectral reflectance of the wings was measured using a spectrograph (SOC710 HS-Portable; Surface Optics Corporation, San Diego, CA, USA). As shown in Fig. 2, the dorsal surfaces of the fore and hind wings were divided into three parts comprising the border, middle, and inside parts. The spectral reflectance was then calculated for the dark and light areas in these three parts. The average spectral reflectance of all the light areas in each part was taken as the spectral reflectance of the light areas in this part, and the average spectral reflectance of all the dark areas in each part was taken as the spectral reflectance of the dark areas in this part. In addition, the excised fore and hind wings were placed under a VHX-1000 super-high magnification lens zoom 3D microscope (VHX-1000; Keyence (China) Co. Ltd., Shanghai, China) to obtain images. The areas of the light areas, dark areas, and the entire wing surface were measured with an X/Y measurement system under the microscope. The ratios of the light areas and dark areas on the wing surfaces were also determined.

## Heat absorption capacities of dark and light areas on the wings

In order to demonstrate that the dark area was responsible for heat absorption via the butterfly wings, we investigated heat absorption in butterflies treated with four different scale removal treatments comprising the removal of scales from dark areas (DSR treatment), the removal of scales from light areas (LSR treatment), the removal of all scales from the wings (ASR treatment), and leaving the scales intact on the wings (IN treatment). Scales are the main tissues for heat absorption of butterfly wings, and the structure color of the wing surface is determined by the scale structure and arrangement. Thus, the

presence or absence of scales can be used to determine the heat absorption ability of a certain area of wing surface. The scales were gently removed from the wing surfaces with a fine brush in each treatment. The treated butterflies were then placed in the dark room. The light intensity was 5,000 lx. After switching the lamp on, during the course of each experiment, we recorded the equilibrated thoracic temperature, rate of thoracic temperature increase, $\Delta T$, and the time required to reach equilibrium. Three female and three male butterflies were tested for each treatment. All of the butterflies used in this experiment were new and unmated five-day old adult *T. limniace*.

## Heat storage and transfer in the wing

We monitored the real-time temperature changes on the wing surface in order to determine the wing surface areas involved with heat storage and transfer during heat absorption by the butterfly wings. The color and markings on the wing surfaces are the same in male and female butterflies, so we only measured the real-time temperature changes on the wing surfaces of male butterflies during heat absorption. In this experiment, five new unmated five-day old male butterflies were examined. A new unmated five-day old male butterfly was fixed to a square white plastic foam board (10 × 10 cm) using insect dissection needles. The wings were fully extended to facilitate real-time temperature monitoring on the wing surface. The butterfly was then placed in the dark room and the light intensity was set to 5,000 lx. The heat absorption rate under 5,000 lx was slower than that at light intensity levels of 15,000–25,000 lx, but it was too slow at 600 lx, so we selected 5,000 lx because it yielded a suitable heat absorption rate for real-time wing temperature monitoring. The real-time temperature changes on the wing surface were recorded using a thermal infrared imager (Flir A600s; FLIR Systems, Inc., Wilson, OR, USA) until the thoracic temperature reached equilibrium.

To determine if dark areas served at heat storage areas, we selected six points in the dark area and six points in the light area and compared their real-time temperature differences. In addition, we selected evenly distributed points from the border of the wing surface to the wing base to identify heat transfer channels. The distributions of all points on the fore and hind wings are shown in Fig. 3A. The distributions of the points on the wing vein on the wing surfaces are shown in Fig. 3B. Five to seven points were selected for each heat transfer channel. Here, we define heat absorption areas as those with fast initial heating rates, but not necessarily high equilibrium temperatures. In contrast, heat storage areas are defined as areas with delayed but rapid heating rates and a high equilibrium temperature.

## Statistical analysis

The interaction effects of the light intensity and wing opening angle on heat absorption were analyzed using two-way ANOVA. We analyzed $\Delta T$, the thoracic equilibrium temperature, rate of thoracic temperature increase, and the time required to reach equilibrium under different light intensities and wing opening angles simultaneously using one-way ANOVA. Duncan's multiple-range test was used to compare multiple means

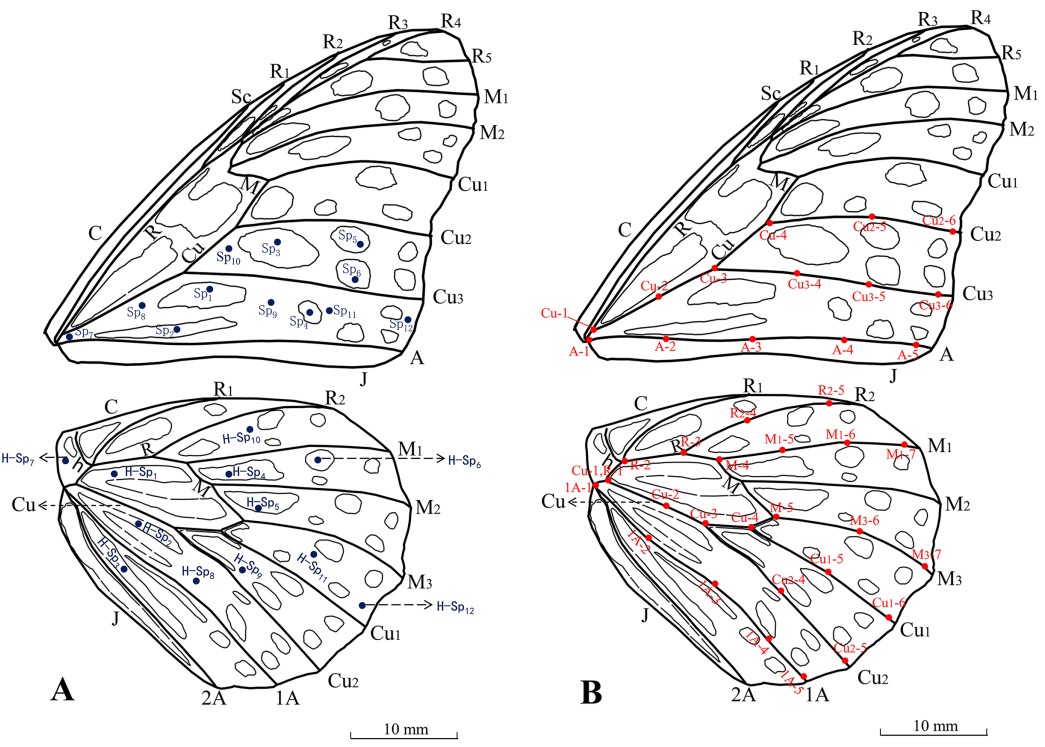

**Figure 3 Distribution of the potential heat storage areas (A) and potential heat transfer channels (B) on the wing surface in adult *Tirumala limniace* butterflies.** In (A), the light gray area represents the heat storage area and the blue points are points used for real-time temperature monitoring in the heat storage areas (fore wing: Sp8–Sp11, hind wing: H-Sp8–H-Sp11), light areas (fore wing: Sp1–Sp6, hind wing: H-Sp1–H-Sp6), wing base (fore wing: Sp7, hind wing: H-Sp7) and non-storage areas (fore wing: Sp12, hind wing: H-Sp12). In (B), the red points are the points used for real-time temperature monitoring during heat transfer in the wing veins. The heat transfer channels are the vein Cu-Cu$_2$ channel, vein Cu-Cu$_3$ channel, and vein A channel in the fore wing, and the vein R-R$_2$ channel, vein R-M-M$_1$ channel, vein Cu-M-M$_3$ channel, vein Cu-Cu$_1$ channel, vein Cu-Cu$_2$ channel, and vein 1A channel in the hind wing.

when significant differences were detected. Differences in ΔT, the thoracic equilibrium temperature, rate of thoracic temperature increase, the time required to reach equilibrium, and the wing expanse between the fore and hind wings were validated using the Student's *t*-test. The differences of spectral reflectance among different areas in male and female wings were analyzed by the Kruskal–Wallis test. The differences in the areas of the light and dark wing surface areas were analyzed using chi-square test. Differences in ΔT, the thoracic equilibrium temperature, rate of thoracic temperature increase, and the time required to reach equilibrium in the SDR, SLR, SAL, and IN treatments were analyzed using one-way ANOVA. Duncan's multiple-range test was used to compare multiple means when significant differences were detected. The trends of real-time temperatures of the monitoring points in the dark areas, light areas, and wing veins were analyzed using time-series ExpDec2 of exponential models $\left(y = A1 * exp\left(\frac{-x}{t1}\right) + A2 * \exp\left(\frac{-x}{t2}\right) + y0\right)$. A1 and A2 is the pre-factor, and they can indicate an increasing trend of the *y* value (the temperature of the points); t1 and t2 are the relaxation time; y0 is the offset, it can reflect the final equilibrium temperature.

**Table 1 Effects of the light intensity and wing opening angle on heat absorption by adult _Tirumala limniace_ butterflies.**

| Sex | Source | | d.f. | Type III SS | Mean square | F | P |
|---|---|---|---|---|---|---|---|
| Male | Equilibrium thoracic temperature | Wing opening angle | 9 | 2,154.7588 | 239.4176 | 6.1 | <0.0001 |
| | | Light intensity | 5 | 44,467.6879 | 8,893.5375 | 226.55 | <0.0001 |
| | | Wing opening angle*Light intensity | 45 | 443.29354 | 9.85097 | 0.25 | 1 |
| | Time required to reach equilibrium temperature | Wing opening angle | 9 | 161.94397 | 17.993775 | 6.91 | <0.0001 |
| | | Light intensity | 5 | 1,165.495037 | 233.099007 | 89.46 | <0.0001 |
| | | Wing opening angle*Light intensity | 45 | 87.682895 | 1.948509 | 0.75 | 0.8659 |
| | ΔT | Wing opening angle | 9 | 2,009.57726 | 223.28636 | 6.16 | <0.0001 |
| | | Light intensity | 5 | 40,432.775 | 8,086.555 | 223.02 | <0.0001 |
| | | Wing opening angle*Light intensity | 45 | 503.31054 | 11.18468 | 0.31 | 1 |
| | Rate of thoracic temperature increase | Wing opening angle | 9 | 2,912.84731 | 323.6497 | 19.99 | <0.0001 |
| | | Light intensity | 5 | 11,774.35845 | 2,354.67169 | 145.42 | <0.0001 |
| | | Wing opening angle*Light intensity | 45 | 3,145.22568 | 69.8939 | 4.32 | <0.0001 |
| Female | Equilibrium thoracic temperature | Wing opening angle | 9 | 1,651.61128 | 183.51236 | 2.59 | 0.0093 |
| | | Light intensity | 5 | 40,823.08502 | 8,164.617 | 15.02 | <0.0001 |
| | | Wing opening angle*Light intensity | 45 | 730.02801 | 16.22284 | 0.23 | 1 |
| | Time to reach equilibrium temperature | Wing opening angle | 9 | 394.8589642 | 43.8732182 | 25.28 | <0.0001 |
| | | Light intensity | 5 | 975.0365768 | 195.0073154 | 112.36 | <0.0001 |
| | | Wing opening angle*Light intensity | 45 | 88.2546565 | 1.9612146 | 1.13 | 0.2964 |
| | ΔT | Wing opening angle | 9 | 1,441.50025 | 160.16669 | 4.28 | <0.0001 |
| | | Light intensity | 5 | 41,421.11186 | 8,284.22237 | 221.13 | <0.0001 |
| | | Wing opening angle*Light intensity | 45 | 815.41275 | 18.12028 | 0.48 | 0.9967 |
| | Rate of thoracic temperature increase | Wing opening angle | 9 | 1,873.31856 | 208.14651 | 11.76 | <0.0001 |
| | | Light intensity | 5 | 10,673.83577 | 2,134.76715 | 120.64 | <0.0001 |
| | | Wing opening angle*Light intensity | 45 | 1,574.69478 | 34.99322 | 1.98 | 0.0018 |

# RESULTS

## Interaction effects of light intensity and wing opening angle on heat absorption by adult butterflies

The wing opening angle and light intensity significantly affected the heat absorption capacity of adult _T. limniace_ butterflies (Table 1). The equilibrium thoracic temperature, time required to reach equilibrium temperature, ΔT, and the rate of thoracic temperature increase were affected significantly by the wing opening angle and light intensity in both male and female butterflies (Table 1). The interaction effect of the light intensity and wing opening angle on the rate of thoracic temperature increase was significant in both male and female butterflies, but the interaction effects on the equilibrium thoracic temperature, time required to reach equilibrium, and ΔT were not significant (Table 1). The direction of such results is described in detail in the following sections.

## Effects of light intensity on the heat absorption capacity

The light intensity significantly affected the heat absorption capacity in adult _T. limniace_ (Tables 2–5). In both the female and male butterflies, within the range of 600–60,000 lx,

**Table 2 Effects of the light intensity and wing opening angle on the equilibrium thoracic temperature (°C) in adult *Tirumala limniace* butterflies.**

| Sex | Wing opening angle | 600 lx | 5,000 lx | 15,000 lx | 25,000 lx | 40,000 lx | 60,000 lx |
|---|---|---|---|---|---|---|---|
| Male | 15° | 30.60 ± 2.47d,A | 47.47 ± 4.78c,A | 61.85 ± 2.64b,A | 64.06 ± 2.33ab,BCDE | 72.26 ± 3.87ab,A | 78.19 ± 8.12a,A |
| | 30° | 32.62 ± 2.01d,A | 48.05 ± 3.00c,A | 61.75 ± 2.44b,A | 63.43 ± 1.81b,CDE | 69.34 ± 2.23b,A | 80.98 ± 6.65a,A |
| | 45° | 34.22 ± 2.72d,A | 52.803 ± 4.29c,A | 63.45 ± 2.99bc,A | 67.96 ± 2.55bc,ABC | 77.26 ± 4.81ab,A | 87.41 ± 9.68a,A |
| | 60° | 36.25 ± 1.73d,A | 54.39 ± 5.48c,A | 63.57 ± 2.12bc,A | 70.69 ± 2.07ab,AB | 80.81 ± 6.56a,A | 81.81 ± 5.69a,A |
| | 75° | 36.28 ± 1.53d,A | 53.34 ± 3.85c,A | 67.06 ± 2.05b,A | 71.79 ± 1.44b,A | 77.52 ± 4.41ab,A | 86.84 ± 5.13a,A |
| | 80° | 35.51 ± 1.73e,A | 48.82 ± 3.57d,A | 63.98 ± 2.44c,A | 67.63 ± 1.51bc,ABC | 76.42 ± 4.39ab,A | 81.78 ± 6.66a,A |
| | 90° | 35.30 ± 0.91e,A | 49.27 ± 4.10d,A | 63.11 ± 3.13c,A | 67.27 ± 3.19bc,ABCD | 75.28 ± 2.94ab,A | 83.51 ± 3.45a,A |
| | 120° | 32.87 ± 1.87d,A | 43.78 ± 1.73c,A | 61.23 ± 2.08b,A | 62.36 ± 1.34b,CDE | 72.93 ± 1.17a,A | 77.19 ± 1.48a,A |
| | 150° | 30.31 ± 2.60d,A | 43.00 ± 2.11c,A | 59.17 ± 1.15b,A | 60.67 ± 2.68b,DE | 68.85 ± 0.98a,A | 73.48 ± 3.16a,A |
| | 180° | 30.43 ± 2.13d,A | 41.35 ± 4.96c,A | 59.55 ± 1.66b,A | 59.47 ± 1.10b,E | 64.21 ± 1.69b,A | 74.19 ± 4.27a,A |
| Female | 15° | 33.52 ± 5.12d,A | 45.10 ± 5.42cd,A | 57.63 ± 7.47bc,A | 64.89 ± 1.50b,AB | 83.30 ± 5.27a,A | 86.27 ± 3.67a,A |
| | 30° | 32.07 ± 4.44d,A | 46.89 ± 5.00cd,A | 59.37 ± 6.11bc,A | 58.90 ± 0.52bc,B | 71.30 ± 0.63ab,BC | 77.09 ± 7.57a,A |
| | 45° | 35.93 ± 5.03d,A | 50.09 ± 5.39cd,A | 62.10 ± 5.85bc,A | 64.56 ± 0.49bc,AB | 76.66 ± 1.29ab,AB | 84.23 ± 8.49a,A |
| | 60° | 34.67 ± 3.96d,A | 48.99 ± 6.29cd,A | 60.34 ± 6.50bc,A | 66.43 ± 0.69abc,A | 74.06 ± 2.34ab,AB | 82.65 ± 11.00a,A |
| | 75° | 35.58 ± 3.65d,A | 49.26 ± 5.81c,A | 60.35 ± 6.09bc,A | 65.64 ± 1.50ab,A | 73.80 ± 1.58ab,AB | 78.40 ± 5.21a,A |
| | 80° | 34.63 ± 3.72d,A | 47.24 ± 3.47cd,A | 59.25 ± 8.07bc,A | 65.10 ± 2.46ab,AB | 72.25 ± 2.24ab,BC | 79.32 ± 6.60a,A |
| | 90° | 35.73 ± 3.51d,A | 46.15 ± 6.78cd,A | 55.56 ± 6.46bc,A | 63.21 ± 2.30ab,AB | 69.86 ± 3.54ab,BC | 74.39 ± 4.54a,A |
| | 120° | 31.95 ± 4.10e,A | 43.89 ± 5.97de,A | 54.64 ± 4.56cd,A | 62.38 ± 1.96bc,AB | 69.66 ± 3.65ab,BC | 75.95 ± 3.63a,A |
| | 150° | 31.01 ± 3.62d,A | 43.53 ± 6.31cd,A | 52.33 ± 7.15bc,A | 58.90 ± 3.08ab,B | 62.78 ± 2.66ab,BC | 71.76 ± 0.62a,A |
| | 180° | 30.80 ± 3.33d,A | 42.39 ± 5.36cd,A | 51.79 ± 5.90bc,A | 58.93 ± 2.11ab,B | 67.12 ± 3.97a,C | 71.81 ± 4.26a,A |

Note:
Different lowercase and capital letters indicate significant differences at *P* < 0.05 in the results obtained at different light intensities and wing opening angles, respectively. Data: mean ± se.

the equilibrium thoracic temperature became significantly higher as the light intensity increased under the same wing opening angle (Table 2). The time required to reach the equilibrium temperature became shorter as the light intensity increased (Table 3). ΔT became significantly higher as the light intensity increased (Table 4). The rate of thoracic temperature increase also became faster as the light intensity increased (Table 5).

However, we also found that when the light intensity was above 40,000 lx, the equilibrium temperature could rapidly exceed 70 °C, which was much higher than the butterflies could tolerate, and thus exposure to a light intensity above 40,000 lx caused the death of the butterflies half an hour after the experiment finished. In summary, a higher light intensity could allow butterflies to absorb more heat (Tables 2–5).

## Effects of wing opening angles on the heat absorption capacity

The wing opening angle also significantly influenced the heat absorption capacity of adult *T. limniace* (Tables 2–5). Compared with wing opening angles 15–45° and 120–180°, when male and female butterflies were at wing opening angles 60–90°, the equilibrium temperatures were higher (Table 2), the times required to reach the equilibrium temperature were much shorter (Table 3), the values of ΔT were much higher (Table 4),

**Table 3 Effects of the light intensity and wing opening angle on the time (min) required to reach equilibrium temperature in adult *Tirumala limniace* butterflies.**

| Sex | Wing opening angle | 600 lx | 5,000 lx | 15,000 lx | 25,000 lx | 40,000 lx | 60,000 lx |
|---|---|---|---|---|---|---|---|
| Male | 15° | 11.97 ± 1.36a,A | 9.68 ± 1.17ab,A | 7.94 ± 0.81b,A | 4.44 ± 0.27c,A | 4.32 ± 0.46c,A | 3.89 ± 0.35c,AB |
| | 30° | 10.60 ± 1.72a,A | 8.64 ± 1.84ab,A | 6.26 ± 0.43bc,AB | 4.12 ± 0.42c,A | 4.36 ± 0.31c,A | 4.67 ± 0.77c,A |
| | 45° | 7.92 ± 1.31a,A | 6.35 ± 0.50ab,A | 4.73 ± 0.20bc,BC | 3.84 ± 0.10c,AB | 3.84 ± 0.65c,A | 3.48 ± 0.71c,ABC |
| | 60° | 9.81 ± 1.25a,A | 6.22 ± 1.11b,A | 3.52 ± 0.53c,C | 3.18 ± 0.08c,BC | 3.18 ± 0.69c,AB | 2.68 ± 0.95c,ABCD |
| | 75° | 8.98 ± 2.04a,A | 6.11 ± 1.45ab,A | 3.25 ± 0.51bc,C | 2.47 ± 0.04c,C | 2.08 ± 0.54c,BC | 1.60 ± 0.40c,CD |
| | 80° | 10.89 ± 1.49a,A | 6.33 ± 1.77b,A | 3.70 ± 0.29bc,C | 2.81 ± 0.18c,C | 2.93 ± 0.52c,ABC | 2.56 ± 0.69c,BCD |
| | 90° | 6.51 ± 0.05a,A | 5.85 ± 0.58a,A | 4.04 ± 0.41b,BC | 2.84 ± 0.27c,C | 1.38 ± 0.10d,C | 1.29 ± 0.14d,D |
| | 120° | 11.62 ± 1.54a,A | 6.24 ± 1.89b,A | 4.67 ± 1.00b,BC | 2.64 ± 0.48b,C | 2.69 ± 0.54b,ABC | 2.64 ± 0.73b,ABCD |
| | 150° | 9.23 ± 0.56a,A | 7.17 ± 1.43a,A | 3.77 ± 0.71b,C | 2.49 ± 0.11b,C | 2.83 ± 0.40b,ABC | 2.95 ± 0.65b,ABCD |
| | 180° | 11.04 ± 1.53a,A | 5.83 ± 1.65b,A | 5.33 ± 1.48b,BC | 3.05 ± 0.25b,BC | 2.13 ± 0.53b,BC | 3.16 ± 0.40b,ABCD |
| Female | 15° | 13.64 ± 1.40a,A | 10.25 ± 0.94b,A | 9.51 ± 1.26b,A | 4.72 ± 0.52c,A | 6.22 ± 0.81c,A | 5.03 ± 0.37c,A |
| | 30° | 12.74 ± 2.15a,A | 10.71 ± 1.52a,A | 8.69 ± 1.29ab,A | 4.12 ± 0.40c,AB | 4.69 ± 0.65bc,B | 4.66 ± 0.79bc,AB |
| | 45° | 8.01 ± 0.41a,B | 6.27 ± 0.71b,B | 4.84 ± 0.21bc,B | 3.21 ± 0.24cd,BC | 3.46 ± 0.50cd,BC | 2.79 ± 0.94d,BC |
| | 60° | 8.18 ± 1.14a,B | 4.77 ± 0.37b,B | 4.23 ± 0.93b,B | 3.02 ± 0.61b,BC | 2.65 ± 0.47b,C | 3.18 ± 1.15b,ABC |
| | 75° | 6.48 ± 0.11a,B | 4.73 ± 0.30b,B | 3.14 ± 0.56c,B | 2.16 ± 0.21cd,C | 2.34 ± 0.09cd,C | 1.46 ± 0.28d,C |
| | 80° | 9.13 ± 0.40a,B | 6.91 ± 0.57ab,B | 4.93 ± 1.68bc,B | 2.92 ± 0.32c,BC | 2.76 ± 0.52c,C | 2.64 ± 0.71c,C |
| | 90° | 8.31 ± 1.10a,B | 4.85 ± 0.66b,B | 3.47 ± 0.66bc,B | 2.83 ± 0.20bc,BC | 1.98 ± 0.50c,C | 1.43 ± 0.11c,C |
| | 120° | 9.26 ± 0.42a,B | 5.37 ± 0.48b,B | 3.84 ± 0.51c,B | 3.21 ± 0.21c,BC | 3.55 ± 0.17c,BC | 2.57 ± 0.49,C |
| | 150° | 9.32 ± 0.46a,B | 6.47 ± 0.32b,B | 3.47 ± 0.35c,B | 3.15 ± 0.34cd,BC | 2.96 ± 0.33cd,C | 2.33 ± 0.14d,C |
| | 180° | 9.12 ± 1.22a,B | 6.67 ± 1.31a,B | 3.76 ± 0.75b,B | 3.55 ± 0.67b,AB | 3.33 ± 0.43b,BC | 2.36 ± 0.35b,C |

**Note:**
Different lowercase and capital letters indicate significant differences at $P < 0.05$ in the results obtained at different light intensities and wing opening angles, respectively. Data: mean ± se.

and he rates of thoracic temperature increase were faster (Table 5). However, the influence of wing opening angle was only seen at some light intensities, particularly the higher light intensities. Thus, the optimal wing opening angles for heat absorption ranged from 60° to 90°.

## Heat absorption capacities of the fore wing and hind wing

The heat absorption capacities of both the fore wing and hind wing were significantly affected by the light intensity where the capacity of the fore wings was higher than that of the hind wings (Table 6). The Student's *t*-test was used to analyze the differences in the heat absorption capacities of the fore wing and hind wing in both female and male butterflies (Table 6). The results showed that the heat absorption capacity of the fore wing was much higher than that of the hind wing in both female and male butterflies (details see Table 6). The equilibrium temperatures and values of ΔT were positively higher for the fore wings than the hind wings under 600–40,000 lx (Table 6), while the times required to reach equilibrium temperature under 600–60,000 lx were significantly shorter in the fore wings than the hind wings, and the rates of thoracic temperature increase were also significantly faster in the fore wings than the hind wings (Table 6). In summary,

**Table 4 Effects of the light intensity and wing opening angle on the thoracic temperature excess (ΔT) (°C) in adult *Tirumala limniace* butterflies.**

| Sex | Wing opening angle | 600 lx | 5,000 lx | 15,000 lx | 25,000 lx | 40,000 lx | 60,000 lx |
|---|---|---|---|---|---|---|---|
| Male | 15° | 9.01 ± 2.01b,B | 21.41 ± 4.01b,A | 37.71 ± 1.57a,BC | 41.07 ± 2.45a,BCD | 48.96 ± 3.53a,ABC | 50.80 ± 8.71a,A |
| | 30° | 10.89 ± 0.95d,AB | 22.78 ± 2.84c,A | 37.17 ± 0.89b,BC | 40.81 ± 2.72b,BCD | 45.79 ± 1.62ab,BC | 54.47 ± 7.13a,A |
| | 45° | 12.72 ± 2.23d,AB | 27.52 ± 3.91cd,A | 39.25 ± 1.60bc,AB | 45.68 ± 2.00ab,ABC | 54.01 ± 3.95ab,AB | 60.34 ± 9.89a,A |
| | 60° | 14.56 ± 1.06d,A | 28.97 ± 5.05c,A | 39.37 ± 0.66bc,AB | 47.58 ± 1.70ab,AB | 57.72 ± 5.74a,A | 55.18 ± 6.36a,A |
| | 75° | 15.15 ± 1.40e,A | 27.84 ± 3.42d,A | 42.39 ± 0.89c,A | 49.25 ± 1.10bc,A | 54.77 ± 3.86ab,AB | 61.35 ± 5.88a,A |
| | 80° | 14.33 ± 1.09c,A | 23.47 ± 3.78c,A | 40.12 ± 1.01b,AB | 45.40 ± 1.36ab,ABCD | 53.87 ± 4.12a,AB | 56.87 ± 7.37a,A |
| | 90° | 14.07 ± 0.14e,A | 25.01 ± 3.60d,A | 38.89 ± 1.73c,ABC | 45.12 ± 3.13bc,ABCD | 52.65 ± 2.55ab,AB | 58.93 ± 3.89a,A |
| | 120° | 11.65 ± 1.70d,AB | 18.68 ± 2.47c,A | 37.27 ± 1.26b,BC | 39.27 ± 1.85b,CD | 49.94 ± 0.41a,ABC | 52.34 ± 1.58a,A |
| | 150° | 8.96 ± 1.52d,B | 18.52 ± 1.62c,A | 35.26 ± 0.62b,C | 38.98 ± 2.36b,CD | 45.20 ± 1.22a,BC | 48.16 ± 3.48a,A |
| | 180° | 10.34 ± 1.95c,AB | 15.36 ± 4.23c,A | 36.91 ± 0.49b,BC | 38.22 ± 2.28b,D | 41.65 ± 2.45ab,C | 48.49 ± 4.22a,A |
| Female | 15° | 10.24 ± 3.12c,A | 21.09 ± 3.81c,A | 33.06 ± 5.95b,A | 43.37 ± 0.72b,A | 60.16 ± 4.40a,A | 62.58 ± 3.01a,A |
| | 30° | 9.03 ± 2.63d,A | 23.27 ± 3.72c,A | 34.91 ± 4.65b,A | 36.35 ± 1.35b,A | 48.04 ± 1.25a,BC | 53.51 ± 5.89a,A |
| | 45° | 12.80 ± 3.10e,A | 25.18 ± 3.54d,A | 37.53 ± 4.31c,A | 42.57 ± 0.93bc,A | 53.81 ± 0.51ab,AB | 59.59 ± 6.94a,A |
| | 60° | 11.70 ± 1.87d,A | 25.03 ± 4.51cd,A | 35.51 ± 4.96bc,A | 44.49 ± 0.82ab,A | 51.30 ± 1.48a,BC | 58.97 ± 8.52a,A |
| | 75° | 12.75 ± 2.00e,A | 25.38 ± 4.17d,A | 35.78 ± 4.52c,A | 43.78 ± 0.82bc,A | 51.18 ± 0.81ab,BC | 55.04 ± 3.50a,A |
| | 80° | 11.16 ± 1.34e,A | 23.37 ± 1.52d,A | 34.66 ± 6.53c,A | 43.84 ± 1.72bc,A | 49.71 ± 1.48ab,BC | 56.05 ± 4.39a,A |
| | 90° | 12.66 ± 2.06d,A | 22.96 ± 4.84cd,A | 31.16 ± 4.79bc,A | 38.83 ± 4.68ab,A | 47.80 ± 2.59a,BCD | 50.24 ± 2.28a,A |
| | 120° | 8.83 ± 1.69e,A | 19.94 ± 4.02d,A | 28.71 ± 3.80c,A | 41.82 ± 1.33b,A | 47.11 ± 2.69ab,BCD | 52.35 ± 2.16a,A |
| | 150° | 9.21 ± 1.90c,A | 20.21 ± 4.04b,A | 27.40 ± 5.23b,A | 38.11 ± 2.12a,A | 40.48 ± 2.06a,D | 47.85 ± 1.97a,A |
| | 180° | 7.62 ± 1.42d,A | 18.72 ± 3.19c,A | 28.16 ± 4.02b,A | 39.17 ± 1.76a,A | 45.71 ± 3.01a,CD | 47.28 ± 2.82a,A |

**Note:**
Different lowercase and capital letters indicate significant differences at $P < 0.05$ in the results obtained at different light intensities and wing opening angles, respectively.
Data: mean ± se.

within the range of 600–60,000 lx, the fore and hind wings could both absorb much more heat at a faster rate as the light intensity increased. The heat absorption capacity of the fore wing was much higher than that of the hind wing (Table 6).

## Identification of heat absorption and non-heat absorption areas

The spectral reflectance of the wings indicated that the dark areas of the wings were used for heat absorption, whereas the light areas did not absorb heat (Fig. 4). The spectral reflectance values of the light areas on both the fore and hind wings were all much higher than those of the dark areas (Male fore wing: $Z = 585.340$, $P < 0.0001$; Male hind wing: $Z = 589.787$, $P < 0.0001$; Female fore wing: $Z = 583.716$, $P < 0.0001$; Female hind wing: $Z = 574.677$, $P < 0.0001$; Fig. 4). The dark areas were significantly larger on both the fore and hind wings than the light areas (male fore wing: $X^2 = 20.841$, $P < 0.0001$; male hind wing: $X^2 = 10.562$, $P = 0.0012$; female fore wing: $X^2 = 16.237$, $P < 0.0001$; female hind wing: $X^2 = 7.242$, $P = 0.0071$). However, the light areas still accounted for about 20% of the fore wing area and about 29% of the hind wing area. In summary, the dark areas could be used to absorb heat, whereas the light areas may prevent the butterfly from absorbing too much heat.

**Table 5 Effects of the light intensity and wing opening angle on the rate (°C/min) of thoracic temperature increase in adult *Tirumala limniace* butterflies.**

| Sex | Wing opening angle | 600 lx | 5,000 lx | 15,000 lx | 25,000 lx | 40,000 lx | 60,000 lx |
|---|---|---|---|---|---|---|---|
| Male | 15° | 0.78 ± 0.19d,E | 2.36 ± 0.67d,A | 4.86 ± 0.59c,D | 9.26 ± 0.46b,D | 11.42 ± 0.42a,CD | 12.87 ± 1.09a,B |
| | 30° | 1.07 ± 0.16e,CDE | 2.94 ± 0.79d,A | 6.01 ± 0.52c,CD | 10.00 ± 0.49b,D | 10.55 ± 0.39ab,D | 11.82 ± 0.54a,B |
| | 45° | 1.63 ± 0.24d,ABC | 4.46 ± 0.92d,A | 8.32 ± 0.50c,BCD | 11.88 ± 0.33b,CD | 14.58 ± 1.66b,CD | 17.72 ± 1.14a,B |
| | 60° | 1.55 ± 0.26d,BCD | 4.96 ± 1.30cd,A | 11.65 ± 1.58bc,AB | 14.98 ± 0.50b,BC | 19.31 ± 2.67ab,CD | 23.93 ± 4.76a,B |
| | 75° | 1.80 ± 0.31d,AB | 5.49 ± 2.09d,A | 13.71 ± 2.12cd,A | 19.98 ± 0.61bc,A | 29.28 ± 5.61b,B | 41.47 ± 6.65a,A |
| | 80° | 1.36 ± 0.20e,BCDE | 4.50 ± 1.78de,A | 10.99 ± 0.99cd,AB | 16.37 ± 1.45bc,AB | 19.12 ± 2.17ab,CD | 24.63 ± 4.40a,B |
| | 90° | 2.16 ± 0.03e,A | 4.49 ± 1.12de,A | 9.82 ± 1.05cd,ABC | 16.16 ± 1.88c,AB | 38.36 ± 2.75b,A | 46.26 ± 3.92a,A |
| | 120° | 1.00 ± 0.05d,DE | 4.07 ± 1.81cd,A | 8.92 ± 2.21bcd,BCD | 15.88 ± 2.78abc,BC | 20.19 ± 4.18ab,BCD | 23.83 ± 7.42a,B |
| | 150° | 0.96 ± 0.12c,DE | 2.92 ± 0.87c,A | 9.97 ± 1.69b,ABC | 15.68 ± 1.13ab,BC | 16.65 ± 2.37a,CD | 17.98 ± 3.74a,B |
| | 180° | 0.94 ± 0.17e,DE | 3.00 ± 1.23de,A | 7.90 ± 1.77cd,BCD | 12.61 ± 0.71bc,BCD | 21.47 ± 4.12a,BC | 15.59 ± 1.00ab,B |
| Female | 15° | 0.77 ± 0.26c,B | 2.14 ± 0.55c,B | 3.44 ± 0.18c,C | 9.40 ± 1.02b,BC | 10.00 ± 1.53ab,C | 12.60 ± 1.25a,C |
| | 30° | 0.72 ± 0.16d,B | 2.33 ± 0.60cd,B | 4.03 ± 0.06c,C | 8.93 ± 0.58b,C | 10.63 ± 1.38ab,BC | 11.75 ± 0.81a,C |
| | 45° | 1.57 ± 0.30d,AB | 4.27 ± 1.13cd,AB | 7.80 ± 1.00bcd,B | 13.45 ± 1.24bc,BC | 16.17 ± 2.19b,BC | 25.35 ± 6.32a,BC |
| | 60° | 1.52 ± 0.37c,AB | 5.18 ± 0.57c,A | 8.82 ± 1.17bc,AB | 16.26 ± 3.79ab,AB | 20.79 ± 4.24a,ABC | 21.32 ± 3.89a,C |
| | 75° | 1.97 ± 0.31d,A | 5.50 ± 1.24cd,A | 11.64 ± 0.82c,A | 20.61 ± 1.99b,A | 21.92 ± 1.01b,AB | 39.78 ± 5.38a,A |
| | 80° | 1.23 ± 0.18c,AB | 3.47 ± 0.53c,AB | 7.77 ± 1.35bc,B | 15.36 ± 1.65ab,ABC | 19.67 ± 4.50a,ABC | 24.23 ± 5.83a,BC |
| | 90° | 1.64 ± 0.45e,AB | 4.69 ± 0.59d,AB | 9.38 ± 1.39c,AB | 14.00 ± 2.36b,BC | 27.93 ± 8.03a,A | 35.74 ± 3.70a,AB |
| | 120° | 0.97 ± 0.23c,B | 3.68 ± 0.57c,AB | 7.50 ± 0.62bc,B | 13.22 ± 1.34b,BC | 13.40 ± 1.42b,BC | 22.46 ± 5.58a,BC |
| | 150° | 0.98 ± 0.19d,B | 3.13 ± 0.65d,AB | 7.92 ± 1.22c,B | 12.46 ± 1.69b,BC | 13.89 ± 0.92b,BC | 20.64 ± 0.93a,C |
| | 180° | 0.86 ± 0.21d,B | 3.30 ± 1.32cd,AB | 7.92 ± 1.48bc,B | 12.14 ± 2.95b,BC | 14.15 ± 1.72b,BC | 20.87 ± 2.81a,C |

**Note:**
Different lowercase and capital letters indicate significant differences at $P < 0.05$ in the results obtained at different light intensities and wing opening angles, respectively. Data: mean ± se.

## Heat absorption capacities of dark and light areas on the wings

Scale removal treatment significantly impacted equilibrium thoracic temperature (Fig. 5). Based on the multirange tests, the effect was due to removal of the dark scales and all scales, while there was no difference in the response variables between intact butterflies and those with light scales removed. When the scales were removed from the light areas of the wings, the equilibrium thoracic temperature, $\Delta T$, the time required to reach equilibrium temperature, and the rate of thoracic temperature increase were not significantly different from those in the butterflies without scale removal. However, their equilibrium thoracic temperatures (male: $F_{3,8} = 28.269$, $P < 0.0001$; female: $F_{3,8} = 21.132$, $P < 0.0001$) and the values of $\Delta T$s (male: $F_{3,8} = 15.686$, $P < 0.0001$; female: $F_{3,8} = 29.17$, $P < 0.0001$) were significantly higher, and the time required to reach the equilibrium temperature (male: $F_{3,8} = 5.96$, $P = 0.0019$; female: $F_{3,8} = 29.170$, $P < 0.0001$) and the rates of thoracic temperature increase (male: $F_{3,8} = 34.839$, $P < 0.0001$; female: $F_{3,8} = 31.613$, $P < 0.0001$) were significantly shorter and faster, respectively, than those of butterflies with all scales removed from the wings or with scales removed from the dark areas (Fig. 5). These results indicate the dark areas of

**Table 6 Heat absorption capacities of the fore wing and hind wing in adult *Tirumala limniace*.**

| Light intensity | | Equilibrium thoracic temperature (°C) | | Time (min) required to reach equilibrium temperature | | $\Delta T$ (°C) | | Rate (°C/min) of thoracic temperature increase | |
|---|---|---|---|---|---|---|---|---|---|
| | | Fore wing | Hind wing | Fore wing | Hind wing | Fore wing | Hind wing | Fore wing | Hind wing |
| Male | 600 lx | $30.99 \pm 0.65^{*}$ | $27.52 \pm 0.88$ | $8.79 \pm 0.39^{*}$ | $11.23 \pm 0.83$ | $11.56 \pm 0.55^{ns}$ | $10.37 \pm 0.75$ | $1.32 \pm 0.06^{*}$ | $0.95 \pm 0.11$ |
| | 5,000 lx | $47.55 \pm 0.67^{**}$ | $40.82 \pm 1.17$ | $5.73 \pm 0.22^{*}$ | $8.43 \pm 0.94$ | $27.12 \pm 0.54^{**}$ | $21.53 \pm 1.20$ | $4.78 \pm 0.25^{**}$ | $2.69 \pm 0.27$ |
| | 15,000 lx | $60.63 \pm 0.73^{**}$ | $50.57 \pm 2.06$ | $4.65 \pm 0.18^{**}$ | $7.93 \pm 0.52$ | $38.96 \pm 0.87^{**}$ | $29.70 \pm 2.31$ | $8.45 \pm 0.42^{**}$ | $3.83 \pm 0.40$ |
| | 25,000 lx | $67.32 \pm 0.52^{**}$ | $62.38 \pm 1.07$ | $4.17 \pm 0.25^{**}$ | $6.28 \pm 0.46$ | $44.49 \pm 0.34^{*}$ | $40.80 \pm 1.31$ | $10.85 \pm 0.59^{**}$ | $6.74 \pm 0.69$ |
| | 40,000 lx | $68.18 \pm 1.02^{**}$ | $60.44 \pm 0.55$ | $3.51 \pm 0.10^{**}$ | $5.96 \pm 0.41$ | $45.23 \pm 0.81^{**}$ | $38.44 \pm 1.12$ | $12.98 \pm 0.55^{**}$ | $6.59 \pm 0.45$ |
| | 60,000 lx | $76.96 \pm 2.88^{ns}$ | $70.55 \pm 0.72$ | $3.65 \pm 0.09^{**}$ | $6.12 \pm 0.52$ | $51.98 \pm 2.77^{ns}$ | $45.97 \pm 0.59$ | $14.25 \pm 0.68^{**}$ | $7.78 \pm 0.63$ |
| Female | 600 lx | $31.38 \pm 0.46^{**}$ | $26.90 \pm 0.90$ | $9.39 \pm 0.21^{*}$ | $11.44 \pm 0.78$ | $13.28 \pm 0.81^{**}$ | $5.90 \pm 0.90$ | $1.41 \pm 0.08^{**}$ | $0.54 \pm 0.10$ |
| | 5,000 lx | $47.72 \pm 1.31^{ns}$ | $42.96 \pm 2.51$ | $8.02 \pm 0.18^{*}$ | $12.23 \pm 1.34$ | $27.49 \pm 1.01^{ns}$ | $21.96 \pm 2.51$ | $3.43 \pm 0.13^{**}$ | $1.80 \pm 0.13$ |
| | 15,000 lx | $61.47 \pm 0.53^{**}$ | $57.82 \pm 0.67$ | $4.98 \pm 0.13^{**}$ | $8.64 \pm 0.22$ | $40.16 \pm 0.75^{**}$ | $34.82 \pm 0.67$ | $8.09 \pm 0.27^{**}$ | $4.05 \pm 0.17$ |
| | 25,000 lx | $65.92 \pm 0.62^{ns}$ | $64.12 \pm 0.60$ | $4.03 \pm 0.17^{**}$ | $8.06 \pm 0.75$ | $43.06 \pm 0.75^{ns}$ | $41.12 \pm 0.60$ | $10.80 \pm 0.52^{**}$ | $5.26 \pm 0.35$ |
| | 40,000 lx | $72.87 \pm 0.46^{**}$ | $67.18 \pm 1.11$ | $3.73 \pm 0.21^{**}$ | $6.58 \pm 0.48$ | $50.43 \pm 0.56^{**}$ | $44.18 \pm 1.11$ | $13.72 \pm 0.71^{**}$ | $6.86 \pm 0.43$ |
| | 60,000 lx | $75.57 \pm 1.05^{ns}$ | $72.84 \pm 1.44$ | $3.74 \pm 0.17^{**}$ | $6.22 \pm 0.33$ | $50.26 \pm 1.19^{ns}$ | $49.84 \pm 1.44$ | $13.54 \pm 0.56^{**}$ | $8.12 \pm 0.47$ |

**Note:**

\*\*, \*, and ns indicate the significance levels of differences in the results (equilibrium thoracic temperature, time required to reach equilibrium temperature, $\Delta T$, and rate of thoracic temperature increase) according to the Student's *t*-test between the fore and hind wings at $P < 0.01$, $P < 0.05$, and $P > 0.05$, respectively. Data: mean ± se.

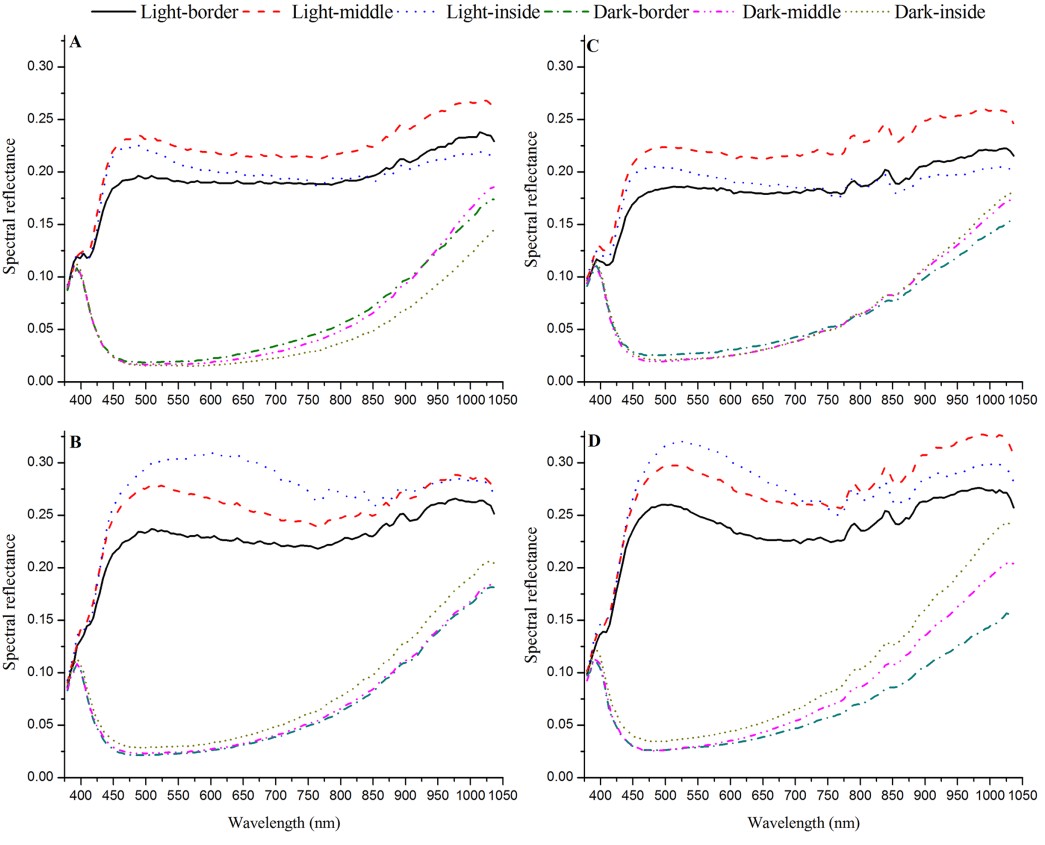

**Figure 4 Spectral reflectance of the male fore wing (A), male hind wing (B), female fore wing (C) and female hind wing (D) in adult *Tirumala limniace* butterflies.**

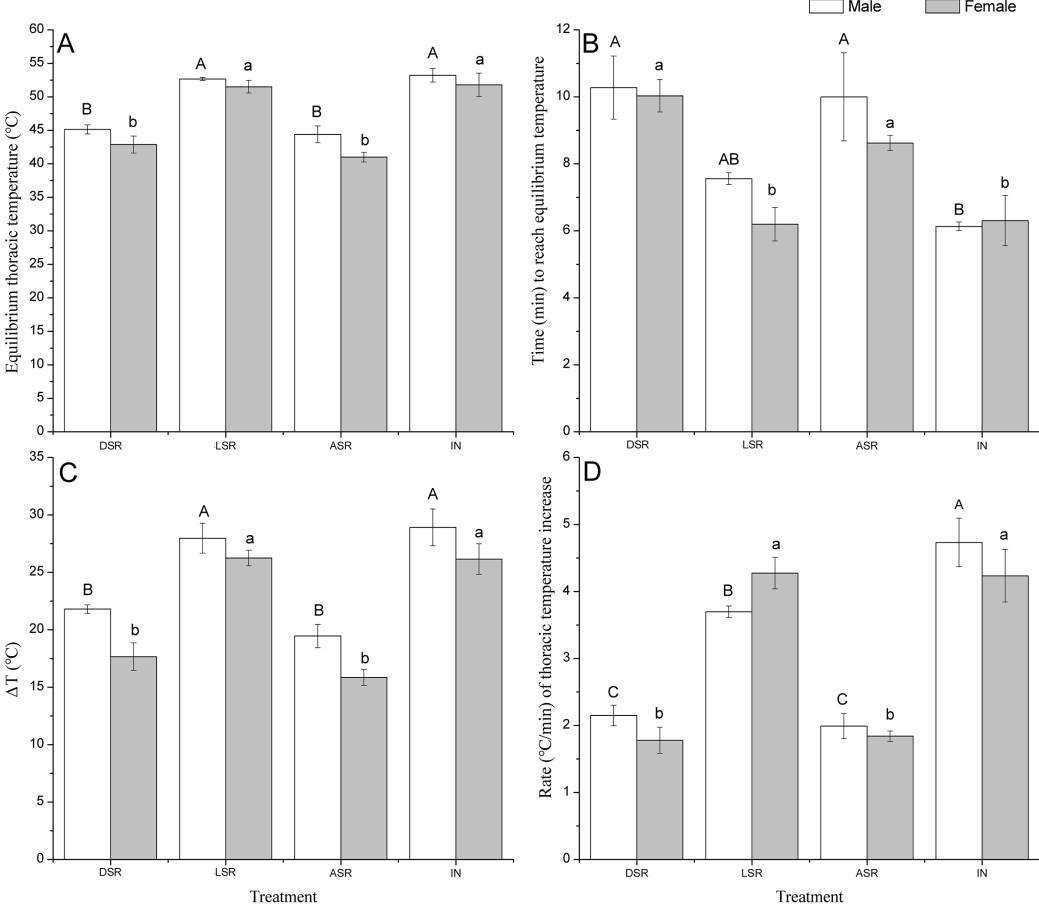

**Figure 5 Equilibrium thoracic temperature (A), time required to reach the equilibrium temperature (B), thoracic temperature excess (ΔT) (C) and rate of thoracic temperature increase (D) in adult *Tirumala limniace* under four scale removal treatments.** DSR treatment: scales removed from dark areas; LSR treatment: scales removed from light areas; ASR treatment: all scales removed from the wing surface; and IN treatment: scales on the wings were left intact. Different lowercase and capital letters indicate significant differences at $P < 0.05$ in the results obtained for males and females among the four scale removal treatments.

the wings are the areas where the butterflies absorb heat, whereas the light areas cannot absorb heat.

## Heat storage and transfer in the wings

### *Real-time temperatures on the wing surface during heat absorption*

The wing temperature increased in *T. limniace* as the lighting time increased, and the temperatures of the wing base and thorax also increased (Fig. 6). After the light was switched on, heat absorption by the wings began immediately. The thoracic temperature reached equilibrium after lighting for 480 s. The temperatures of the light areas on both the fore and hind wings were lower than those of the dark areas during heat absorption in adult *T. limniace* (Fig. 7). As the lighting period continued, heat appeared to accumulate in some areas of the wing surface and the temperature was higher than that of other areas (Figs. 6 and 7).

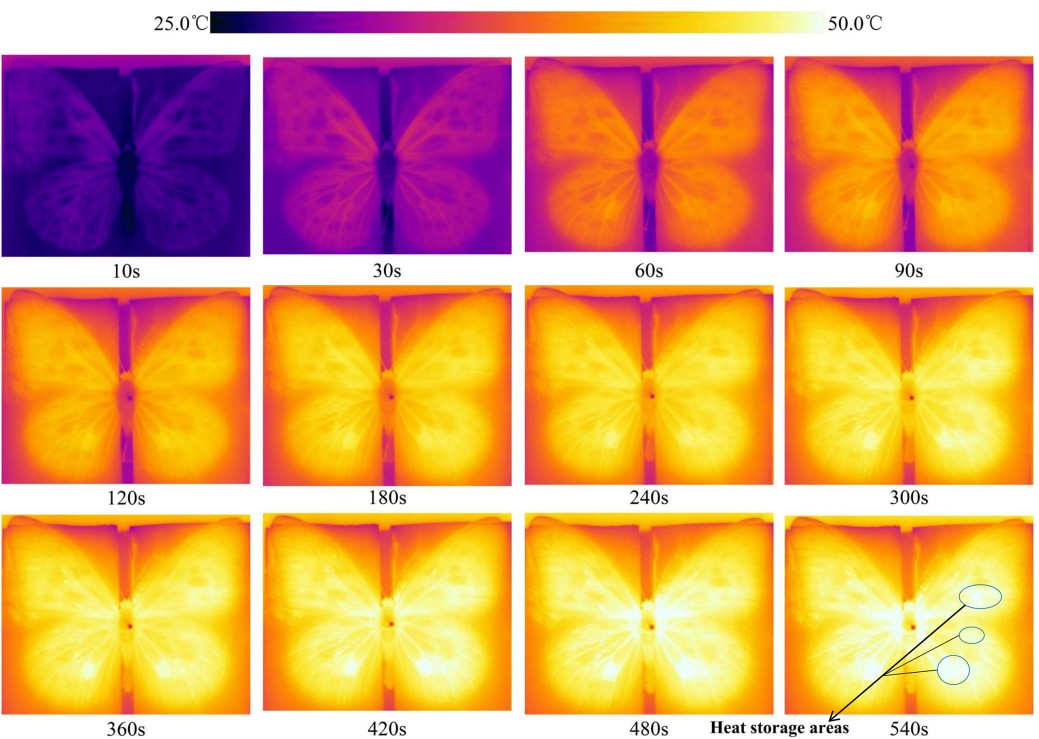

**Figure 6 Real-time temperature on the wing surface during heat absorption by adult *Tirumala limniace* butterflies.** The numbers below each subgraph indicate the monitoring time in seconds.

### Real-time temperatures in the heat storage and non-storage areas on the wing surface during heat absorption

The temperature at each point on the fore and hind wing surfaces increased as the lighting period increased as the analysis of time-series ExpDec2 of exponential models (Fig. 7; Table 7). On the fore wing, the base of the wing (Sp7) heated at a conspicuously slow rate initially (A1), then heated at a higher rate than other points at later times (A2). This suggests that the base of the wing does not absorb heat but is a heat storage area, as further supported by the high equilibrium temperature parameter (y0). The points in the dark area heated at a high, similar rate initially (A1 of Sp8–Sp12), followed by points not in the border (Sp8–Sp11) continuing to heat at a high rate, while the border point (Sp12) showed a decrease in later heating, suggesting it is poor at heat storage (A2 values).The points in the light area always heated at a similar rate with the points in the dark area, but the equilibrium temperatures were lower than the non-border dark areas (y0s of the points Sp1–Sp6 were significantly lower than the points Sp8–Sp11: $t_8 = 2.864$, $P = 0.021$), while being similar to those of the border.

In the hind wing, the results were qualitatively the same as for the forewing, with the base of the wing heating at a conspicuously slow rate initially, followed by a very high late heating rate and high equilibrium value. The non-border dark areas had high heating rates throughout the trial, while the border dark area showed a slow heating rate late in the trial. In contrast to the forewing, we found the equilibrium parameters for the light areas

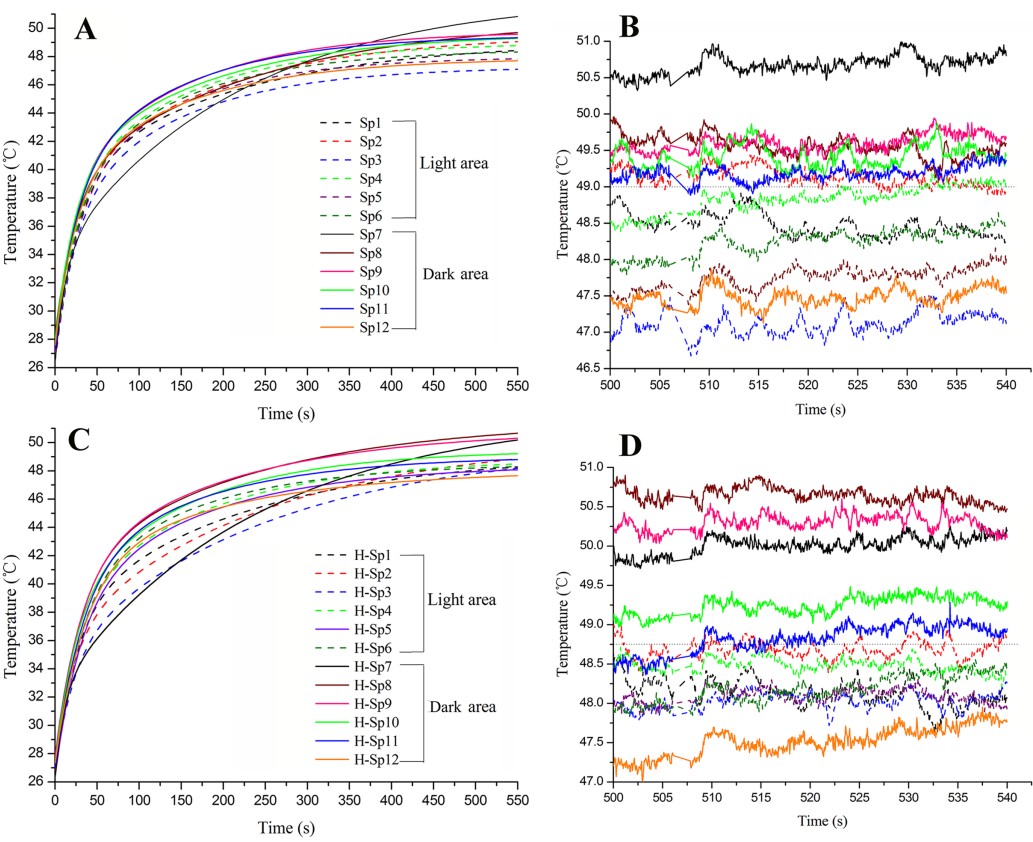

**Figure 7 Real-time temperature at the monitoring points in the heat storage areas and non-heat storage areas during heat absorption by the wing surfaces in adult *Tirumala limniace* butterflies.** (A) Fore wing; (B) real-time temperatures at monitoring points on the fore wing during lighting for 500–540 s; (C) hind wing; (D) real-time temperatures at monitoring points on the hind wing during lighting for 500–540 s.

near the base of the wing (y0 of H-Sp1, H-Sp2 and H-Sp3) were similar to those of the non-border dark areas (H-Sp8–H-Sp11; $t_5 = 0.683$, $P = 0.525$). The possible reason is that these points were close to the base of the wing and were affected by the heat of the base of the wing. The temperatures of other three points in the light area were lower than the non-border dark areas (y0: $t_5 = 2.986$, $P = 0.031$). In summary, for both wings, the non-border dark areas could temporarily store heat and then transfer it to the base of the wing. The dark area of the border could absorb heat from the light source but it could not store the heat, similar to most light areas on the wing (Fig. 7).

### Real-time temperatures in the heat transfer channel in the wing veins during heat absorption

Both in the fore and hind wings, the temperatures of all the veins increased as the lighting time increased as the analysis of time-series ExpDec2 of exponential models (Figs. 8 and 9). The heat rate of the points in all veins failed to show a certain rule according to the distance from the base of the wing (Figs. 8 and 9). But, in all vein channels, the closer points get to the base of the wing, the higher the final equilibrium temperature was (Figs. 8 and 9), and the value of y0 was higher (Table 8). These indicated that with the heat

**Table 7 The results of time-series ExpDec2 of exponential models analysis for trends of real-time temperatures of the monitoring points in the heat storage areas and non-heat storages in butterflies *Tirumala limniace*.**

| Wing type | Area | Point | y0 | A1 | t1 | A2 | t2 | r | F | P |
|---|---|---|---|---|---|---|---|---|---|---|
| Fore wing | Light area | Sp1 | 49.10929 | −12.4412 | 27.33323 | −9.97643 | 204.0954 | 0.99816 | 1.74E + 08 | <0.0001 |
| | | Sp2 | 49.57089 | −11.0479 | 179.7561 | −11.9106 | 24.73968 | 0.99842 | 1.96E + 08 | <0.0001 |
| | | Sp3 | 47.28997 | −10.1232 | 24.05404 | −10.6675 | 137.0862 | 0.99856 | 2.35E + 08 | <0.0001 |
| | | Sp4 | 48.98872 | −10.933 | 140.0638 | −11.428 | 23.81826 | 0.99847 | 2.13E + 08 | <0.0001 |
| | | Sp5 | 47.99227 | −10.6245 | 129.5564 | −10.7854 | 25.69463 | 0.99846 | 2.14E + 08 | <0.0001 |
| | | Sp6 | 48.50339 | −11.4961 | 26.27294 | −10.2551 | 137.9374 | 0.99842 | 2.09E + 08 | <0.0001 |
| | Wing base | Sp7 | 52.29589 | −7.40247 | 12.81485 | −18.4286 | 217.0853 | 0.99897 | 2.03E + 08 | <0.0001 |
| | Non-border dark areas | Sp8 | 50.75721 | −11.7911 | 228.043 | −10.7115 | 24.1697 | 0.99702 | 1.09E + 08 | <0.0001 |
| | | Sp9 | 49.82083 | −11.158 | 143.3776 | −11.5553 | 21.9387 | 0.99868 | 2.56E + 08 | <0.0001 |
| | | Sp10 | 49.65997 | −10.0475 | 166.3924 | −11.7129 | 24.33015 | 0.9976 | 1.50E + 08 | <0.0001 |
| | | Sp11 | 49.53145 | −11.7152 | 24.88213 | −10.5312 | 138.5937 | 0.99852 | 2.28E + 08 | <0.0001 |
| | Border in dark area | Sp12 | 47.91015 | −11.173 | 24.91333 | −9.14257 | 145.2361 | 0.99726 | 1.43E + 08 | <0.0001 |
| Hind wing | Light area | H-Sp1 | 49.17633 | −10.4439 | 25.37711 | −11.6125 | 215.0843 | 0.99741 | 1.18E + 08 | <0.0001 |
| | | H-Sp2 | 50.26874 | −14.1702 | 238.9041 | −8.59803 | 22.34626 | 0.99829 | 1.59E + 08 | <0.0001 |
| | | H-Sp3 | 49.76717 | −7.90298 | 21.82562 | −15.0295 | 244.5349 | 0.99901 | 2.50E + 08 | <0.0001 |
| | | H-Sp4 | 48.85711 | −11.3694 | 25.41496 | −10.9769 | 161.4886 | 0.99852 | 2.08E + 08 | <0.0001 |
| | | H-Sp5 | 48.40098 | −10.3218 | 159.2493 | −11.6644 | 29.78131 | 0.99848 | 1.97E + 08 | <0.0001 |
| | | H-Sp6 | 48.54843 | −12.3569 | 31.24274 | −9.28688 | 151.8395 | 0.9983 | 1.89E + 08 | <0.0001 |
| | Wing base | H-Sp7 | 52.09365 | −6.07772 | 11.27207 | −19.64 | 235.8899 | 0.99925 | 2.53E + 08 | <0.0001 |
| | Non-border dark areas | H-Sp8 | 51.53298 | −10.8477 | 218.4568 | −13.9014 | 28.09526 | 0.99883 | 2.47E + 08 | <0.0001 |
| | | H-Sp9 | 50.90148 | −13.0178 | 30.27843 | −10.1396 | 194.0823 | 0.99808 | 1.60E + 08 | <0.0001 |
| | | H-Sp10 | 49.46966 | −10.9644 | 24.55755 | −11.233 | 145.7819 | 0.99839 | 2.01E + 08 | <0.0001 |
| | | H-Sp11 | 49.06514 | −12.9741 | 31.49138 | −9.03949 | 155.2494 | 0.9979 | 1.53E + 08 | <0.0001 |
| | Border in dark area | H-Sp12 | 48.04232 | −13.2487 | 34.84247 | −7.43888 | 185.9954 | 0.99798 | 1.74E + 08 | <0.0001 |

Note:
ExpDec2 Model: $y = A1 * exp\left(\frac{-x}{t1}\right) + A2 * exp\left(\frac{-x}{t2}\right) + y0$.

transfer in the vein, the closer the points were to the wing base, the higher the temperatures (Figs. 8 and 9).

## DISCUSSION

### Effects of the light intensity on heat absorption

We found that the light intensity significantly affected the heat absorption capacity of the butterfly *T. limniace*. Within the light intensity range from 600 to 60,000 lx, as the light intensity increased, the thoracic equilibrium temperature and ΔT increased significantly, the time to reach the equilibrium temperature was significantly shorter, and the rate of thoracic temperature increase was significantly faster. Our previous studies also showed that compared with other light intensities (243, 860, and 1,280 lx), the equilibrium thoracic temperature of adults exposed to 2,240 lx was higher and the time required to reach it was shorter. In addition, the value of ΔT was higher and the rate of thoracic temperature increase was achieved more quickly (*Liao et al., 2017*). These results suggest that in an appropriate range, a high light intensity can allow adult

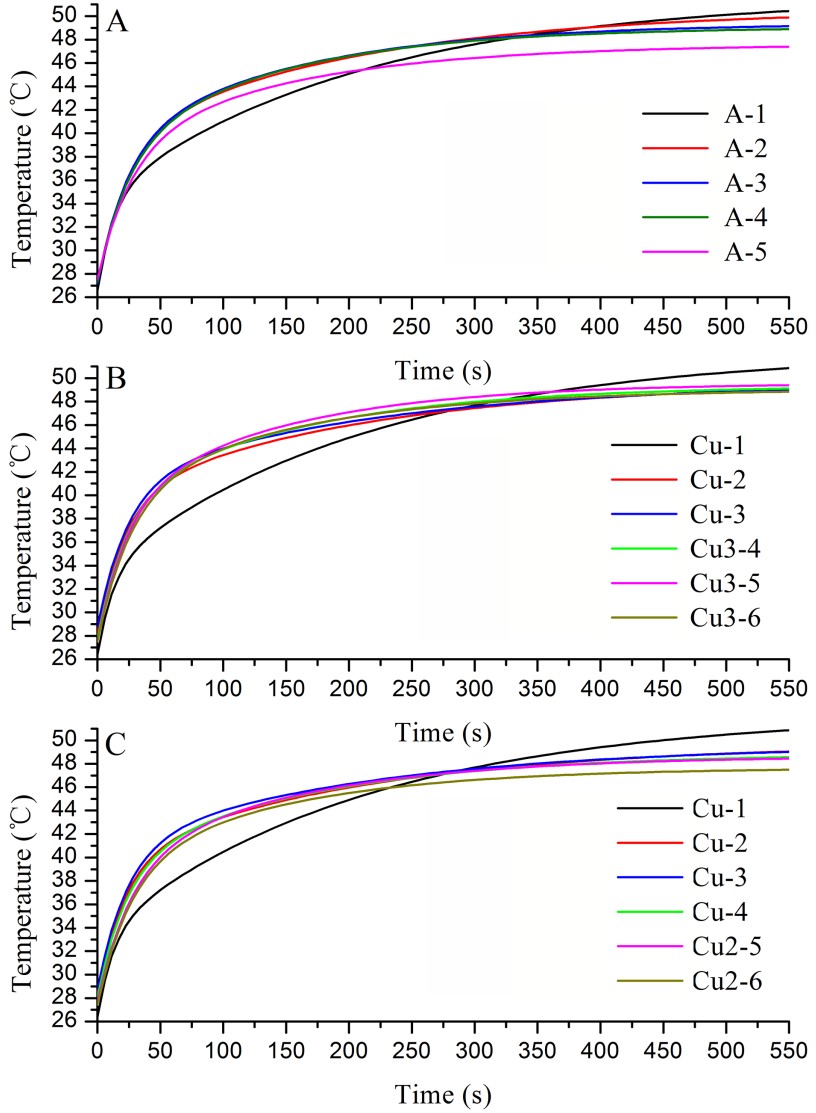

**Figure 8 Real-time temperatures at monitoring points on the heat transfer channels on the fore wing during heat absorption by adult *Tirumala limniace* butterflies.** (A) Vein A channel; (B) vein Cu-Cu₃ channel; (C) vein Cu-Cu₂ channel.

*T. limniace* butterflies to rapidly absorb more heat to elevate their thoracic temperature compared with low light intensities. Similar results were obtained for the butterfly *Heteronympha merope* where its body temperature excess also increased as the level of solar radiation increased (*Barton, Porter & Kearney, 2014*).

The heat absorbed by butterflies is used mainly for autonomous flight and it is directly involved with reproductive behavior (*Shreeve, 1992*; *Kemp & Krockenberger, 2004*; *Bennett, Smith & Betts, 2012*; *Bonebrake et al., 2014*; *Kleckova, Konvicka & Klecka, 2014*; *Mattila, 2015*; *Shanks et al., 2015*; *Han et al., 2016*; *Niu et al., 2016*). In *Bicyclus anynana*, active males increase their own likelihood of copulation and active females increase their likelihood of being courted (*Westerman, Drucker & Monteiro, 2014*). The flight activity of *T. limniace* butterflies is higher under a high light intensity compared with those

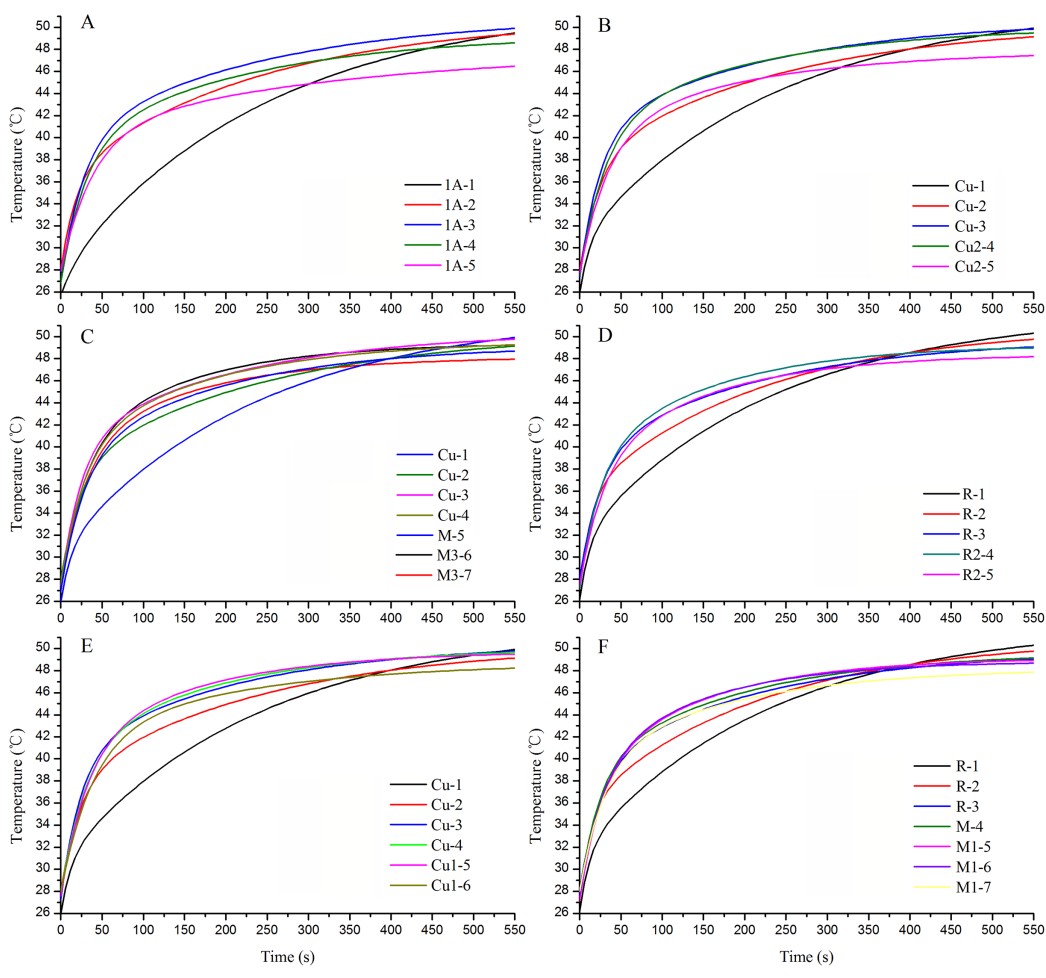

**Figure 9 Real-time temperatures at monitoring points on the heat transfer channels on the hind wing during heat absorption by adult *Tirumala limniace* butterflies.** (A) Vein 1A channel; (B) vein Cu-Cu$_2$ channel; (C) vein Cu-M-M$_3$ channel; (D) vein R-R$_2$ channel; (E) vein Cu-Cu$_1$ channel; (F) vein R-M-M$_1$ channel.

exposed to a low light intensity (*Liao et al., 2017*). The high thorax temperature recorded in *Junonia villida* is probably linked to its high flight speed (*Nève & Hall, 2016*). We found that a higher light intensity increased the heat absorbed by *T. limniace* adults from light sources. Thus, within an appropriate light intensity range, a higher light intensity can allow *T. limniace* adults to absorb more heat to facilitate autonomous flight and increase the flight activity, thereby enhancing the likelihood of reproduction.

We also found that when the light intensity was above 40,000 lx, the equilibrium temperature could rapidly exceed 70 °C, which was much higher than the butterflies could tolerate, and thus exposure to a light intensity above 40,000 lx caused the death of the butterflies half an hour after the experiment finished. In the summer and autumn season of Yuanjiang county, Yunan province, China, the peak of flight activity of butterfly *T. limniace* mainly appeared at 9.00–12.00 and 15.00–17.00 (*Chen et al., 2008*). During these time periods, the sunlight intensity is lower than 40,000 lx, which allows the butterfly to absorb the right amount of heat for autonomous flight. More than 40,000 lx light

**Table 8 The results of time-series ExpDec2 of exponential model analysis for trends of real-time temperatures of the monitoring points in the heat transfer channels of butterflies *Tirumala limniace*.**

| Wing type | Heat transfer channel | Point | y0 | A1 | t1 | A2 | t2 | r | F | P |
|---|---|---|---|---|---|---|---|---|---|---|
| Fore wing | Vein A channel | A-1 | 51.65031 | −7.84003 | 12.75891 | −17.2268 | 207.2903 | 0.99856 | 1.60E + 08 | <0.0001 |
| | | A-2 | 50.5731 | −11.4145 | 195.1354 | −11.8801 | 24.10951 | 0.99828 | 1.81E + 08 | <0.0001 |
| | | A-3 | 49.41572 | −11.5431 | 23.41794 | −10.7117 | 148.3088 | 0.99858 | 2.39E + 08 | <0.0001 |
| | | A-4 | 49.06832 | −10.6785 | 24.03947 | −11.0778 | 133.0077 | 0.9985 | 2.25E + 08 | <0.0001 |
| | | A-5 | 47.6016 | −10.7138 | 26.48365 | −9.33575 | 144.2478 | 0.9974 | 1.45E + 08 | <0.0001 |
| | Vein Cu-Cu$_3$ channel | Cu-1 | 52.26994 | −18.9101 | 211.9113 | −6.97851 | 12.25504 | 0.99928 | 2.77E + 08 | <0.0001 |
| | | Cu-2 | 49.67318 | −11.2014 | 21.28217 | −10.1931 | 197.2287 | 0.99303 | 5.58E + 07 | <0.0001 |
| | | Cu-3 | 49.61431 | −8.88343 | 204.3245 | −11.9366 | 23.56786 | 0.99061 | 4.53E + 07 | <0.0001 |
| | | Cu$_3$-4 | 49.38674 | −11.2699 | 22.10509 | −10.4084 | 150.2459 | 0.99763 | 1.55E + 08 | <0.0001 |
| | | Cu$_3$-5 | 49.59448 | −11.1955 | 22.80552 | −10.8877 | 135.7532 | 0.99861 | 2.52E + 08 | <0.0001 |
| | | Cu$_3$-6 | 49.06385 | −11.857 | 25.909 | −9.68216 | 144.6714 | 0.99801 | 1.80E + 08 | <0.0001 |
| | Vein Cu-Cu$_2$ channel | Cu-1 | 52.26994 | −18.9101 | 211.9113 | −6.97851 | 12.25504 | 0.99928 | 2.77E + 08 | <0.0001 |
| | | Cu-2 | 49.67318 | −11.2014 | 21.28217 | −10.1931 | 197.2287 | 0.99303 | 5.58E + 07 | <0.0001 |
| | | Cu-3 | 49.61431 | −8.88343 | 204.3245 | −11.9366 | 23.56786 | 0.99061 | 4.53E + 07 | <0.0001 |
| | | Cu-4 | 48.88576 | −11.0924 | 23.04441 | −9.71285 | 160.9608 | 0.9957 | 9.08E + 07 | <0.0001 |
| | | Cu$_2$-5 | 48.67759 | −9.81106 | 147.3665 | −11.1861 | 26.23936 | 0.9984 | 2.25E + 08 | <0.0001 |
| | | Cu$_2$-6 | 47.66185 | −9.15046 | 138.1042 | −11.0852 | 25.98615 | 0.99752 | 1.55E + 08 | <0.0001 |
| Hind wing | Vein A channel | 1A-1 | 52.1982 | −24.3625 | 250.3032 | −2.16858 | 17.72072 | 0.99953 | 2.62E + 08 | <0.0001 |
| | | 1A-2 | 50.81314 | −14.4636 | 236.0123 | −8.44482 | 17.70421 | 0.99773 | 1.26E + 08 | <0.0001 |
| | | 1A-3 | 50.97916 | −12.6379 | 26.86096 | −11.4056 | 233.0519 | 0.99857 | 2.06E + 08 | <0.0001 |
| | | 1A-4 | 49.37377 | −10.3706 | 212.6243 | −12.156 | 29.12618 | 0.99859 | 2.17E + 08 | <0.0001 |
| | | 1A-5 | 47.77004 | −12.3352 | 38.22446 | −7.54459 | 313.1892 | 0.9978 | 1.71E + 08 | <0.0001 |
| | Vein Cu-Cu$_2$ channel | Cu-1 | 52.06394 | −4.74979 | 11.58034 | −21.4292 | 239.0546 | 0.99951 | 3.26E + 08 | <0.0001 |
| | | Cu-2 | 50.40223 | −9.85654 | 22.84531 | −12.667 | 238.1548 | 0.99708 | 1.05E + 08 | <0.0001 |
| | | Cu-3 | 50.69774 | −12.9494 | 23.92048 | −10.4392 | 219.3316 | 0.99674 | 1.04E + 08 | <0.0001 |
| | | Cu$_2$-4 | 50.08737 | −13.0645 | 30.0606 | −9.55433 | 198.0817 | 0.99787 | 1.51E + 08 | <0.0001 |
| | | Cu$_2$-5 | 47.87697 | −12.3765 | 33.8684 | −7.79342 | 191.6719 | 0.99745 | 1.42E + 08 | <0.0001 |
| | Vein Cu-Cu$_1$ channel | Cu-1 | 52.06394 | −4.74979 | 11.58034 | −21.4292 | 239.0546 | 0.99951 | 3.26E + 08 | <0.0001 |
| | | Cu-2 | 50.40223 | −9.85654 | 22.84531 | −12.667 | 238.1548 | 0.99708 | 1.05E + 08 | <0.0001 |
| | | Cu-3 | 50.45283 | −12.7617 | 23.48126 | −10.4624 | 201.8555 | 0.99669 | 1.02E + 08 | <0.0001 |
| | | Cu-4 | 50.10416 | −12.5917 | 28.82723 | −9.83279 | 178.2873 | 0.99704 | 1.10E + 08 | <0.0001 |
| | | Cu$_1$-5 | 49.79113 | −13.0948 | 31.51791 | −9.12978 | 159.7369 | 0.99766 | 1.39E + 08 | <0.0001 |
| | | Cu$_1$-6 | 48.74334 | −13.6109 | 37.68257 | −7.2046 | 208.5855 | 0.9979 | 1.65E + 08 | <0.0001 |
| | Vein Cu-M-M$_3$ channel | Cu-1 | 49.43767 | −11.7424 | 28.44878 | −10.0544 | 141.542 | 0.99771 | 1.46E + 08 | <0.0001 |
| | | Cu-2 | 48.1947 | −11.6759 | 29.62228 | −8.88782 | 150.9702 | 0.99744 | 1.41E + 08 | <0.0001 |
| | | Cu-3 | 52.06394 | −4.74979 | 11.58034 | −21.4292 | 239.0546 | 0.99951 | 3.26E + 08 | <0.0001 |
| | | Cu-4 | 50.40223 | −9.85654 | 22.84531 | −12.667 | 238.1548 | 0.99708 | 1.05E + 08 | <0.0001 |
| | | M-5 | 50.45283 | −12.7617 | 23.48126 | −10.4624 | 201.8555 | 0.99669 | 1.02E + 08 | <0.0001 |
| | | M$_3$-6 | 49.65581 | −12.0016 | 27.14308 | −10.1177 | 170.5792 | 0.9977 | 1.44E + 08 | <0.0001 |
| | | M$_3$-7 | 49.23426 | −10.7417 | 184.3841 | −11.596 | 26.7489 | 0.99845 | 2.00E + 08 | <0.0001 |

| Wing type | Heat transfer channel | Point | y0 | A1 | t1 | A2 | t2 | r | F | P |
|---|---|---|---|---|---|---|---|---|---|---|
| | Vein R-M-M$_1$ channel | R-1 | 52.24029 | −20.6188 | 232.3034 | −5.47058 | 11.76307 | 0.99934 | 2.63E + 08 | <0.0001 |
| | | R-2 | 50.9655 | −8.43454 | 13.48904 | −15.4319 | 215.4063 | 0.99783 | 1.24E + 08 | <0.0001 |
| | | R-3 | 49.8506 | −10.9777 | 207.8621 | −10.8279 | 24.80101 | 0.99655 | 9.79E + 07 | <0.0001 |
| | | M-4 | 49.69803 | −10.7669 | 184.4267 | −11.2514 | 23.31607 | 0.99702 | 1.16E + 08 | <0.0001 |
| | | M$_1$-5 | 49.16783 | −11.2546 | 26.17524 | −10.719 | 143.3344 | 0.99783 | 1.50E + 08 | <0.0001 |
| | | M$_1$-6 | 48.86012 | −10.2357 | 135.2823 | −10.939 | 26.87515 | 0.99784 | 1.61E + 08 | <0.0001 |
| | | M1-7 | 48.2474 | −11.5281 | 29.4377 | −8.72021 | 176.2904 | 0.99685 | 1.19E + 08 | <0.0001 |
| | Vein R-R$_2$ channel | R-1 | 52.24029 | −20.6188 | 232.3034 | −5.47058 | 11.76307 | 0.99934 | 2.63E + 08 | <0.0001 |
| | | R-2 | 50.9655 | −8.43454 | 13.48904 | −15.4319 | 215.4063 | 0.99783 | 1.24E + 08 | <0.0001 |
| | | R-3 | 49.8506 | −10.9777 | 207.8621 | −10.8279 | 24.80101 | 0.99655 | 9.79E + 07 | <0.0001 |
| | | R$_2$-4 | 49.27588 | −11.1911 | 23.79768 | −10.8977 | 151.1524 | 0.99836 | 2.04E + 08 | <0.0001 |
| | | R$_2$-5 | 48.42841 | −10.3834 | 27.89841 | −10.5232 | 146.3984 | 0.99825 | 1.89E + 08 | <0.0001 |

**Note:**
ExpDec2 Model: $y = A1 * \exp\left(\frac{-x}{t1}\right) + A2 * \exp\left(\frac{-x}{t2}\right) + y0$.

intensity could make the butterflies absorb more heat than the body can tolerate, burning the body and causing death. Some studies indicated that butterflies can adjust their basking posture to reduce heat absorption (*Rawlins, 1980*; *Tsuji, Kingsolver & Watt, 1986*), for example, closing the wings and turning the body opposite the sunlight. In *T. limniace*, in the field, we observed that when the sunlight was above 40,000 lx, the butterflies would fly into the woods and perch on branches to avoid the sunlight.

## Effects of the wing opening angle on heat absorption

The wing opening angle also significantly influenced the heat absorption capacity of *T. limniace* adults. Some previous studies have shown that butterflies with fully open wings have the highest heat absorption capacity (*Tsuji, Kingsolver & Watt, 1986*; *Berwaerts et al., 2001*; *De Keyser et al., 2015*). When the butterfly *Polyommatus icarus* fully opens its wings, the warming rates are maximized and its body temperature is also highest (*De Keyser et al., 2015*). *Pararge aegeria* butterflies with fully opened wings are able to absorb more heat and cool faster compared with those with half-opened wings (*Berwaerts et al., 2001*). Other studies suggest that the heat absorption capacity of butterflies is highest in the optimal range for the wing opening angle (*Kingsolver & Moffat, 1982*; *Barton, Porter & Kearney, 2014*). For example, in *Pieris*, normal and black base manipulation of butterflies can achieve maximum temperature excesses at wing angles of 30–40°, whereas the optimal wing angles for the highest temperature excess in black distal treatment butterflies are 60–90° (*Kingsolver, 1987*). Thus, regardless of whether the wings are normal or melanized, heat absorption by butterflies requires a certain optimal wing opening angle that can vary depending on the butterfly species. We found that the optimal wing opening angle range for heat absorption by *T. limniace* was 60–90°. In this range, the equilibrium temperatures were significantly higher, the times required to reach the equilibrium temperature were much shorter, the values of ΔT were

higher, and the rates of thoracic temperature increase were significantly faster. Thus, an appropriate wing opening angle can enhance the heat absorption capacity of butterflies. In the wild, when butterflies need heat for autonomous flight, adult *T. limniace* don't fully open their wings; however, they would make the wings maintain a certain angle. Thus, wing opening angles 60–90° are suitable for heat absorption of the butterfly *T. limniace*.

## Differences in heat absorption between the fore and hind wings

Our results showed that the heat absorption capacity of the fore wings was significantly higher than that of the hind wings. The equilibrium temperatures and values of $\Delta$Ts were significantly higher for the fore wings than the hind wings in males exposed to light at 600–40,000 lx and females exposed to light at 600, 15,000, and 40,000 lx. The times required to reach equilibrium temperature and the rates of thoracic temperature increase were significantly shorter and faster, respectively, in the fore wings than those in the hind wing for both males and females exposed to light at 600–60,000 lx. These results suggest that the fore wing is the major tissue used for heat absorption in butterflies. Thus, the wing size significantly impacts the heat absorption capacity of butterflies (*Kammer & Bracchi, 1973*; *Berwaerts et al., 2001*; *Berwaerts, Van Dyck & Aerts, 2002*). Some studies suggest that butterflies with larger wings can absorb more heat than those with smaller wings (*Heinrich, 1986*; *Kingsolver, 1987*; *Schmitz, 1994*). For example, a study of 20 Australian butterflies showed that those with the highest wing loadings had the highest thorax temperature at take-off (*Nève & Hall, 2016*). Therefore, the fore wings allow *T. limniace* butterflies to absorb more heat than the hind wings under the same lighting conditions. However, in *Pararge aegeria*, butterflies with larger wings heated up more slowly compared with those with small wings (*Berwaerts et al., 2001*).

## Identification of heat absorption and non-heat absorption areas

The spectral reflectance from light areas on both the fore and hind wings was much higher than that from dark areas in *T. limniace* adults. Thus, the absorbance and thermal absorptivity of the dark areas was significantly greater than that of the light areas. Color can affect an animal's temperature because dark surfaces absorb more solar energy than light surfaces and the energy is converted into heat (*Stuart-Fox, Newton & Clusella-Trullas, 2017*). Previous studies indicate that wing color can limit the heat absorption capacity, where darker coloration can absorb more heat from sunlight or other light sources than lighter coloration (*Watt, 1968*; *Kingsolver, 1987*, *1988*; *Berwaerts et al., 2001*). For example, darker *Pararge aegeria* butterflies have a much higher warming rate (*Van Dyck, Matthysen & Dhondt, 1997*). We found that when the scales were removed from the light areas on the wings, the equilibrium thoracic temperature and value of $\Delta$Ts were significantly higher, and the time required to reach the equilibrium temperature and the rate of thoracic temperature increase were significantly shorter and faster, respectively, than those in butterflies with all the scales removed from the wings or with scales removed from the dark areas. Thus, the dark areas of the wings absorb heat in the butterfly *T. limniace* rather than the light areas.

Some studies suggest that melanization of the wing surfaces can promote the absorption of heat from sunlight in butterflies (*Kingsolver & Wiernasz, 1991*; *Stoehr & Wojan, 2016*). The butterfly *Parnassius phoebus* lives at high altitudes and high latitudes, and it uses melanization to obtain heat from solar radiation, where the color of its wings becomes darker as the altitude and latitude increase (*Guppy, 1986*). Melanization inside the wing surfaces can improve the heat absorption capacity of butterflies whereas melanization at the distal wing surfaces cannot affect heat absorption (*Kammer & Bracchi, 1973*; *Kingsolver, 1987*; *Brashears, Aiello & Seymoure, 2016*). *Wasserthal (1975)* showed that most of the heat transferred from the wing to the body originates from the 15% of the wing surface nearest to the body in the butterflies *Papilio machaon*, *Apatura ilia*, and *Papilio troilux*. Thus, the dark areas of the wing surface near the body are the main heat absorption areas in butterflies. Our results are similar to those obtained in previous studies. In *T. limniace*, we found that the temperatures were markedly higher in the dark areas in the mid-posterior region near the wing base in wing cells $A$-$CU_3$ and $Cu_2$-$Cu_3$ in the fore wings, as well as in wing cells $1A$-$Cu_2$, $Cu_1$-$Cu_2$, $M_3$-$Cu_1$, and $R_2$-$M_1$ in the hind wings compared with other wing areas during lighting for 300–540 s.

The color of a butterfly wing mainly comprises iridescence and structural colors (*Nijhout, 1991*; *Michielsen, De Raedt & Stavenga, 2009*; *Han et al., 2016*; *Siddique et al., 2017*). The part of the wing with a structural color is used to absorb heat from the sun (*Bosi et al., 2008*). In *Pararge aegeria*, normal butterflies reach a higher equilibrium thoracic temperature than descaled butterflies, probably because the wings without scales absorb less radiation (*Berwaerts et al., 2001*). In the present study, we found that removing the scales from the dark areas of the wings significantly reduced the heat absorption capacity of butterflies, whereas removing the scales from the light areas did not affect normal heat absorption. Therefore, the dark areas on the dorsal wing surfaces in *T. limniace* may mainly comprise structural colors.

High reflectance can reduce the absorption of heat from sunlight (*Stuart-Fox, Newton & Clusella-Trullas, 2017*). Thus, light colors are not conducive to heat absorption by butterflies. We found that removing the scales from the light areas on the wing surfaces did not affect the absorption of heat by butterflies. In *Anartia fatima*, it was previously shown that blackening the white bands on the wing surface did not affect its equilibrium temperature, but adding white bands decreased the rate of heating (*Brashears, Aiello & Seymoure, 2016*). Thus, the light areas on the wing surfaces may prevent butterflies from absorbing heat excessively quickly and increasing the body temperature to an intolerable level. We also found that light areas accounted for about 20% of the fore wing area and about 29% of the hind wing area. However, some studies indicate that light areas have high spectral reflectance and they could be used as information for intraspecies identification or for detecting the opposite sex in butterflies (*Emmel, 1972*; *Taylor, 1973*; *Seymoure & Aiello, 2015*). The role of the light areas on butterfly wings in heat absorption will be addressed in our future research.

## Heat storage and transfer in the wing

In the butterfly *T. limniace*, the temperatures of the dark areas were markedly higher in the mid-posterior region near the wing base in wing cells $A$-$Cu_3$ and $Cu_2$-$Cu_3$ on the fore

wings, as well as in wing cells $1A$-$Cu_2$, $Cu_1$-$Cu_2$, $M_3$-$Cu_1$, and $R_2$-$M_1$ on the hind wings compared with the other wing areas during lighting for 300–540 s. Hence, the heat absorbed by the wings of the butterfly *T. limniace* needs to be temporarily stored in heat storage areas on the wing surface, before it is transferred to the wing base and thorax to elevate the thoracic temperature above the ambient level to trigger autonomous flight. Thus, the dark areas in the mid-posterior region near the wing base in wing cells $A$-$Cu_3$ and $Cu_2$-$Cu_3$ in the fore wings, and in wing cells $1A$-$Cu_2$, $Cu_1$-$Cu_2$, $M_3$-$Cu_1$, and $R_2$-$M_1$ in the hind wings were temporary heat storage areas on the wing surface. The heat was transferred from the heat storage areas to the wing base and thorax through the veins near the storage areas. During the heat transfer process, the temperatures of the wing base and thorax were the same as the heat storage areas and much higher than those of other areas of the wing surfaces on both the fore and hind wings, and the thoracic temperature reached equilibrium after lighting for 480 s. Thus, the channels for heat transfer were the veins near the storage areas.

*Kingsolver (1985)* suggested that the wings act as solar reflectors in *Pieris* butterflies to reflect solar radiation onto the body in order to increase its temperature. However, some studies have shown that the absorption of heat by butterfly wings depends mainly on the internal structure of the scale itself, where the structure called a "photonic crystal" can convert the absorbed light into heat for autonomous flight (*Li et al., 2004*; *Han et al., 2013*, *2015a*, *2015b*; *Luohong, 2014*). In the butterfly *Trogonoptera brookiana*, the scales have longitudinal ridges that run through the scales and the surfaces of the scales comprises a set of raised longitudinal quasiparallel lamellae (ridges), where the spaces between adjacent ridges are filled with a netlike reticulum comprising pores (*Han et al., 2015a*). This "photonic crystal" structure in the scales may help the wings to absorb heat. Therefore, the heat absorbed is not reflected to the wing base and thorax through the wing surface, but instead it is transmitted via the internal tissue in the wing. Our results showed that the heat was transferred from the heat storage areas to the wing base and thorax through veins $Cu_2$, $Cu_3$, $Cu$, and $A$ in the fore wings, and veins $1A$, $Cu_2$, $Cu_1$, $Cu$, $M_1$, $M_3$, $M$, $R_2$, and $R$ in the hind wings during lighting for 0–540 s. This suggests that the heat absorbed by the wings is transferred from the heat storage areas on the wing surface to the wing base and thorax through wing veins.

## CONCLUSIONS

In this study, we showed that in the optimal light intensity range, a higher light intensity could help butterflies to absorb more heat, and the optimal wing opening angles for heat absorption ranged from 60° to 90°. The heat absorption capacity of the fore wing was much greater than that of the hind wing. The dark areas on the wing surfaces were the areas where heat was absorbed by the butterfly wing. The dark areas in the mid-posterior region near the wing base in wing cells $A$-$Cu_3$ and $Cu_2$-$Cu_3$ in the fore wing, and in wing cells $1A$-$Cu_2$, $Cu_1$-$Cu_2$, $M_3$-$Cu_1$, and $R_2$-$M_1$ in the hind wings were temporary heat storage areas on the wing surface. We speculate that the heat was transferred from heat storage areas to the wing base and thorax through veins $Cu_2$, $Cu_3$, $Cu$, and $A$ in the fore wings, and veins $1A$, $Cu_2$, $Cu_1$, $Cu$, $M_1$, $M_3$, $M$, $R_2$, and $R$ in the hind wings during

lighting for 0–540s. In the future, we will investigate the heat storage mechanism in butterfly wings, as well the heat transfer mechanism in the wing veins and the role of light areas on the wings in heat absorption.

## ACKNOWLEDGEMENTS

We would like to thank our reviewers and the editor of *PeerJ* for their comments, which have improved the manuscript tremendously. We also thank Chuanjing Liu for her assistance in original materials. We thank International Science Editing for language editing.

### Funding

This study was sponsored by the National Natural Science Foundation of China (NSFC 31702072), the Fundamental Research Funds for the Central Non-profit Research Institution of CAF (CAFYBB2017QA012), and the Special Fund for Forest Scientific Research in the Public Welfare of China (201504305). There was no additional external funding received for this study. The funders had no role in study design, data collection and analysis, decision to publish, or preparation of the manuscript.

### Grant Disclosures

The following grant information was disclosed by the authors:
National Natural Science Foundation of China: NSFC 31702072.
Fundamental Research Funds for the Central Non-profit Research Institution of CAF: CAFYBB2017QA012.
Special Fund for Forest Scientific Research in the Public Welfare of China: 201504305.

### Competing Interests

The authors declare that they have no competing interests.

### Author Contributions

- Huaijian Liao conceived and designed the experiments, performed the experiments, analyzed the data, contributed reagents/materials/analysis tools, prepared figures and/or tables, authored or reviewed drafts of the paper, approved the final draft.
- Ting Du performed the experiments, authored or reviewed drafts of the paper, approved the final draft.
- Yuqi Zhang performed the experiments, authored or reviewed drafts of the paper, approved the final draft.
- Lei Shi conceived and designed the experiments, performed the experiments, analyzed the data, contributed reagents/materials/analysis tools, prepared figures and/or tables, authored or reviewed drafts of the paper, approved the final draft.
- Xiyu Huai performed the experiments, authored or reviewed drafts of the paper, approved the final draft.
- Chengli Zhou performed the experiments, approved the final draft.
- Jiang Deng performed the experiments, approved the final draft.

## Data Availability

Raw data is available in the Supplemental Materials.

## Supplemental Information

Supplemental information for this article can be found online at http://dx.doi.org/10.7717/peerj.6648#supplemental-information.

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
