# Peer review of "Capacity for heat absorption by the wings of the butterfly Tirumala limniace (Cramer)"

_PeerJ, doi:10.7717/peerj.6648_

## Round 0.1 · original submission · Major Revisions

I have received two reviews for this paper. Both reviewers were impressed by the large amount of data presented and the value of the information for the field. I agree with this assessment, and feel the data represent a very impressive collection of studies. However, the reviewers also recommended major revisions that will need to be completed before the ms can be considered further. I also note that one of the reviewers provides detailed annotations to the pdf.

The reviewer comments that I highlight are:

1. Adding statistical analyses of the comparisons presented in Figs 12-14.
2. Putting the vast majority of the statistical results within the data tables (or relying on the significance groupings when sufficient). The results become too difficult to follow with so many test statistics in them
3. Summarizing the text of the results when similarities exist between sexes and light intensities. Refer the reader to the statistics in tables for the details.
4. The labelling on Fig 4 is difficult to see, particularly the blue text on grey. In addition, the light-area lines are unclear in the legend of Fig 12.
5. The ecological relevance of butterflies finding 60,000lx stressful.

My own editorial comments:
1. Write with more intuitive explanation with respect to the wing geography – cells, veins and sampling points (eg “distal section of the forewing”). This will make it clearer for readers not familiar with wing nomenclature.
2. Figure 2 and Table 1 can be omitted. Figs 6 & 7 can be combined; Figs 9 & 10 can be combined.
3. How was the heating rate calculated? Total change in temp over time, or fitting a curve to the heating data?
4. Before the discussion, it is not clear why scale removal might be a sensible manipulation. Please provide motivation earlier in the ms.
5. The discussion has more repetition of the results than in necessary.

Reviewer 1 ·

Basic reporting

The article is written in clear professional English and is clearly understandable throughout. The background of the study is properly introduced and the literature cited fits the broader field of the study. A formal analysis of butterfly thermoregulation is currently lacking in literature, as well as a quantification of the contribution of dark wing areas to the thermoregulatory process. Hence, the article clearly contributes to fill a gap in literature. However, in its current state the article has some remarkable flaws in the way results are presented making it unsuitable for publication.

Experimental design

Methods are described in sufficient detail. However, in some cases it is not completely clear whether the butterflies used were alive or dead. Hence, more details should be provided regarding the condition of the butterflies tested.

Validity of the findings

The structure of the article mostly conforms to the standards of scientific publications, but there are few remarkable exceptions: the second half of the results section, which is also the one including the focal findings of this paper, is built upon data inferred from figures rather than based on a statistical analysis formally quantifying the effects asserted. Based on the raw data provided, the authors have the data for a thorough statistical analysis, but this was not performed. For example, the fact that the so called heat storage areas heat more than other wing areas should be statistically tested, and figure 11 should only serve as an example. For all the data referring to figures 12 to 14, the authors should perform a time-series analysis, or at least compare the slopes of the different curves during the linear phase, and compare the final temperatures after they plateau. A statistical analysis formally supporting the findings is essential for publication.
Another weakness of the paper is the lack of references to the ecological relevance of the measurements assayed. First of all, it is not clear (at least to me) why this butterfly is a good model organism to study heat absorption. In addition, the authors test light intensities ranging from 600 to 60 000 lx, even though all butterflies die at intensities higher than 40 000 lx. This result should be extensively discussed in the light of the natural conditions the butterfly faces in the wild. Does this mean that butterflies do not bask during the warmest times of the day because of lethal light intensity? Or could it be that the halogen lamps impact the butterflies more heavily than natural conditions? I encourage the authors to discuss these points as critically relevant to the focus of the paper.
Conclusions are well stated and linked to the research question, but they are flawed to the extent that they are partially based on results not supported by an adequate statistical analysis. But this issue will disappear once the authors provide adequate statistical tests for their findings.
Finally, there was little or no attempt to link the findings to the natural basking and thermoregulatory behaviors of the butterfly (e.g. are the optimal wing opening angles found here observed more commonly by naturally basking butterflies?).

Annotated reviews are not available for download in order to protect the identity of reviewers who chose to remain anonymous.

Reviewer 2 ·

Basic reporting

Expression could be revised for minor grammar points. The results are horrendously complex and laboriously presented - some sections are almost impenetrable for the repetition of raw statistical tests.

Experimental design

See general comments.

Validity of the findings

See general comments.

Additional comments

I was interested in this manuscript in terms of how it seeks to define exactly how heat exchange occurs in butterflies. The question is of general curiosity for workers in Lepidoptera. The focus is heavily mechanistic; therefore the title may better reflect this. Overall there are useful insights here, but the presentation is horrendously complex and laborious. The results section “Effects of light intensity on the heat absorption capacity” is simply unreadable. Very few readers could be expected to extract the principal findings. I strongly suggest that this material be better incorporated in Figures or Tables. The authors have obviously invested an enormous amount of work in this study, which is highly commendable. The presentation however may not generate the highest or broadest impact because it is highly specific to the taxon, but it is worthy of publication in my view.

Minor:
Line 34 (and throughout): It is not just flight that body temperature determines in these insects – indeed it is their greater metabolic rate, and hence all metabolic processes (in females this may determine the rate of ovarian development, for example).

Line 46: the extensive literature on “reflectance basking” by Kingsolver and colleagues is mentioned later, which is great, but it could perhaps be also stated here.

“Light intensity” should be simply referred to as such, and not “The light intensity”

Line 271: The term “double-factorial analysis of variance” is unclear if not unconventional. The statistics should be phrased in terms of general linear modelling, which more broadly encompasses what used to be referred to as one-way & factorial ANOVA, etc.

The study relies on light intensity in lux as the basic unit of illumination. Lux is a measure of intensity that is “calibrated” for human perception. It would be unequivocal to use this under various sunlight conditions, but it may deviate from heat gain under artificial lighting. The authors should recognize this caveat in their argument.

---

## Round 0.2 · Major Revisions

The authors have performed a major revision and I have received a re-review by one of the original reviewers. The revision is a great improvement in terms of data presentation. The paper presents a lot of interesting research, but I agree with the reviewer that there are still a large number of aspects of the paper that are unclear or so long as to make it difficult to draw meaning from the results. I have provided my own review as well in the hopes that the next revision is substantial and suitable for publication. Please address these comments fully before resubmitting.

The sections on time series (starting line 406) need substantial revision. 1), Tables 10-12 can support comparisons of heating rate (A1 and A2), but cannot be drawn on to comment on temperature itself. If you want to comment on temperature itself, refer only to the figures. Moreover, the F and p values are not terribly useful, as they only support the fact that the curve fits the data. 2) the wording in this section needs to be revised substantially to refer to intuitive sections of the wing rather than only the location code. It is extremely difficult for the reader to draw meaning when constantly flipping between text, tables, time series figure and the wing diagram. As an example, Line 408 – The broad interpretation of the curves is interesting, but should be modified to say that the base of the wing (Sp7) heated at a conspicuously slow rate initially (A1), then heated at a higher rate than other points at later times (A2). Follow this type of revision through all time series analyses. 3) how does Table 10 support that this point was the hottest at the end? Does y0 represent the asymptote in this shape of the function? 4) this section should be shortened to highlight the major pattern rather than detailing each point. This was only 1 animal, so there is little point describing the data in great detail. 5) the occasional inclusion of statististics looking at equilibrium temperature is confusing. What is your replication? How do you have 20+ records and what are you comparing? I suggest removing these. More useful for the heat absorption and storage argument would be the reviewer’s suggestion to perform a t-test on the relevant parameter values (y0, A1, A2?) for the light vs dark samples in each wing. It is noticeable that the A1 of light areas is not lower than that of dark areas – why do you think light areas are heating as fast early on?

In line with the reviewer’s comment, the use of the terms ‘heat absorption area’ ‘heat storage area’ and ‘heat transfer channel’ seems premature before you’ve presented and interpreted the results. Some minor wording changes could fix this. In addition, the distinction between wing heating and thoracic heating should be more precise.
Heading line 199 – ‘Reflectance of light and dark areas’
Line 218 – delete. I suggest only using the terms ‘heat absorption area’ and ‘non-heat absorption area’ in the discussion. It is much simply for the reader and more conservative scientifically to refer to them as light and dark area through the methods, results and figure legends. Whether they are heat absorption areas or not is a matter of interpretation of the results.
Line 226 – “…responsible for thoracic heating via the butterfly wings,….”
Line 253 – delete, as above, ‘heat storage and transfer areas’ would be better inferred in the results or in the discussion
Line 255 – Change to something like: ‘To determine if dark areas served at heat storage areas, we selected 6 points in the dark area and 6 points in the light area…… In addition, we selected evenly distributed points from the border of the wing surface to the wing base to identify heat transfer channels.”
Line 225 vs line 240 – I don’t see the distinction between ‘heat absorption areas’ and ‘heat storage areas’. If there is a real distinction between these two, please clarify in the methods. I note that at line 420 in the results, you seem to be operationally defining heat absorption as the initial fast heating rate (A1) and heat storage as the second slow heating rate (A2). If this is your definition, or one within the field, it should be clarified in the methods.
Line 285 – define all of the parameters in the model. I am not familiar with this function, but based on my investigation of it, Y is not a slope – in fact, it is just the Y value (temperature). Y0 is the offset (intercept). I believe you are actually interested A1 and A2, which are the fast and slow portions of the 2-part exponential function.
Line 297 – what is the basis for saying the impact of light intensity is greater? I suggest removing this as both factors are clearly very important.
Line 316 – specify that the influence of wing angle was only seen at some light intensities, particularly the higher light intensities.
Section starting line 358 – the one-way ANOVAs apply to all treatments, not just to the dark scale removals. To present the statistical results (currently line 363 on) within the text, you will need an overall sentence such as scale removal treatment significantly impacted equilibrium thoracic temperature (stats here), and etc. Then say that, based on the multirange tests, that the effect was due to removal of the dark scales and all scales, while there was no difference in the response variables between intact butterflies and those with light scales removed.
Line 388 and Fig 8 – as these regions will be unfamiliar to the reader, it would help if you provided circles or arrows on one of the panels of Fig 8 to identify the areas you are highlighting the text.
Line 394 – it is not clear to the reader that heat is being transferred from heat storage areas through veins! Change to ‘….it appeared that heat was transferred’. Lines 397-403 should be omitted from this section, as visual inspection Fig 8 is not adequate support for these statements. If they are true, they should be in the nest section, where the data are analysed.
Table 2 and others – for the letters to denote groups – do these come from the two-way ANOVA or the separate one-way ANOVAs? I assume the one-way ANOVAs.
Tables 2-8 – these results are not really the most interesting part of your paper, and having so many tables is unnecessary and distracting from the importance of the results. If you prefer to present all of the means in tables rather than figures, please put the statistical results of Table 6-8 into the data tables (2-5). This can be done by adding a row beneath the ‘male’ section and a row beneath the ‘female’ section of each data table, and a column on the right of each table. The rows contain F(df), p AND SAMPLE SIZE for the ANOVAs comparing wing angles at each light intensity. The columns contain the same statistics for the ANOVAs comparing among light intensities at each wing angle.
Fig. 3. Change the legend to read ‘potential’ heat storage areas and heat transfer channels.
Fig 8. Replace the panel lettering with the time interval
Fig 3. Lettering indicating the different areas does not show well, mostly in the dark section. I would suggest removing the dark grey stippling. As noted by the reviewer, what is the difference between the light areas and non-storage areas? Is a heat storage area defined by your results or an a priori expectation?
Table 9 – where does the lettering for light intensity come from? The methods do not mention an ANOVA. I don’t think these ANOVA are particularly useful in this particular experiment as they are fairly repetitive of the previous section. I suggest deleting the results (Line 326-lin 332) and the table letters so that the reader can focus on the contrast between fore and hind wing.
Delete Fig 6 and place the values for proportion dark in the results text with the test statistics.
Fig 9 axis and legend – change SDR to DSR and change SLR to LSR. These will match ASR.

Reviewer 1 ·

Basic reporting

no comment

Experimental design

no comment

Validity of the findings

no comment

Additional comments

The authors made significant improvements on their paper. However, I still have some concerns about clarity overall. Indeed, the paper provides a massive amount of data, but it is still not presented in a clear and accessible form. One critical point of the paper is to compare the thermal activity of dark and light wing portions, but this difference is not always explicitly tested. Statistical tests, including the time-series, are often based on differences before and after lighting within the same wing portion, rather than specifically comparing cumulatively “dark areas” versus all “light areas”. If the authors could frame the manuscript more clearly around this objective, the paper would read more easily. The results on single wing portions are difficult to follow, but I believe that a comparison between dark areas vs light areas as a whole, especially in the time series analysis, is needed in order to have a clear idea of the significance of the findings. Please find below more detailed comments and suggestions to improve clarity.

1. Are heat storage / non-storage and absorption / non-absorption equivalent terms? Please clarify and choose only one term and use it consistently. Also, the same areas also roughly correspond to dark and light areas, right? Please explain in the very beginning of the methods the relationship (or expected relationship) between dark/light, storage/non-storage, absorption/non-absorption and then choose one term and use it consistently for the whole paper, tables figures and figure captions.

2. Table 6 and 7 could be merged as they are the same, only for male and female. The same for table 11 and 12.

3. Please add a column specifying whether a certain area is considered or not a heat storage area. There is no way a reader will know that by heart when reading the table 10, and it makes the understanding extremely difficult.

4. Please provide what all the letters in table 10 mean. Apparently y0 is important, but there is no way of knowing what it refers to.

5. I appreciate that the authors took the time to provide statistical support for their findings and perform a time-series analysis. However, the statistical support is still missing for the “identification of heat-absorption and non-absorption areas”, only shown in figure 5. Please provide statistical testing. Additionally, it seems that what was assessed in the time series analysis was the difference between the initial and equilibrium temperature within every single wing portion. This information is extremely redundant, and the actual use of the analysis, where all the p-values are extremely highly significant, is unclear. If the point here is only to demonstrate that the initial temperature was lower than the equilibrium temperature after lighting, the same information could have been obtained by measuring three random points on each wing. It is unclear how the authors draw their conclusions about the role of heat storage and non-storage areas, and I have a feeling that y0 is used for this purpose? What is really interesting but not tested here is the difference in the heat absorption ability (the slope of how fast heat is absorbed) of light vs dark wing areas. Indeed, when I suggested to perform a time-series analysis I meant to perform separate analyses for heat-storage and non-storage areas and then compare statistically whether they differ significantly. That is, perform a statistical model testing whether the heat absorption slope significantly differ on whether it is on a heat storage or non-heat storage area, similar to what done with the scale removal experiment. This is very relevant information.
Furthermore, it is not entirely clear how the authors conclude that the heat it transferred from one portion to the other, and how they identify how heat moves across the channels. Again, I believe this is based on the values of y0, but it cannot be understood from the text and the authors should provide more details in the methods so that the reader can understand and interpret the findings.


Minor comments:

L 15: “The scientific question remains unclear”. The scientific question is clear, but the answer remains unclear, I guess?

L16: Please include 1-2 sentences giving some background on why thus butterfly species was used to answer this question.

L15-21: This sentence is way too long and detailed for an abstract. Try to state more concisely, for example by saying “We measured a number of wing traits relevant for heat absorption including X, Y, Z” where X, Y and Z are the ones the authors found being most relevant.

L27: “Cu2, Cu3, Cu, and A in the fore wing, and veins 1A, Cu2, Cu1, Cu, M1, M3, M, R2, and R “. Again, the authors can expect no general reader to know what these symbols mean in the wing topography. This is way too much detail for an abstract and should be left out at this stage. The newly added part explaining in words what this means is clear and informative enough.

L 100-105: The fact of flying on sunny days and not on cloudy and rainy days is common to all butterfly species. What I meant by adding some background on the study organism was to explain why this particular butterfly species, Tirumala limniace, was a good model to study these questions. Is it something about the coloration of the wings, or the patterning that makes it a better model than other butterfly species occurring in the same climate? In other words, can the authors explain why this particular species was chosen instead of any another butterfly species occurring at the same latitude?
Also: replace “when on the sunny day” with “on sunny days” and “in field” with “in the field”.

L198: Please add unit of measure for thoracic temperature increase.

L293: “Sp8, Sp9, Sp10, and Sp11”: so these are dark wing portions? Please correct this in figure 9, where every portion is labelled as “light”.

L318-325: If I understood correct, what is reported in tables 10-12 is basically a comparison between the temperature of a certain wing portion at time 0 and its temperature at the final time point of the heating range, where the heating curve is fitted to have an exponential distribution? Is it so? So these data are basically summarizing what shown in figure 8, which is nice. However, I believe that the most relevant analysis for the scope of this paper is a comparison between the time series of heat storage areas and non-heat storage areas.

Also, if y is the slope, what does y0 represent? And how about A1, A2, t1, t2 and X? Please provide more details on the model variables.

L329-338: This section as it stands is not very informative. Saying that something affected something else gives no information on the direction of such effect. Please be more informative by replacing “significantly” with “positively” or “negatively”, or using phrases as “the time required to reach equilibrium temperature was shorter for X wing angles and Y light intensities”.

L340: As above. Replace “significantly” with “positively”, so that the reader does not need to go through six tables to understand the general pattern.

L384: I do not understand what expanse means. Is it the length of the wing? If so, this is a rather common characteristic in butterflies, and does not provide new information, especially in the context of heat absorption capacity. Please explain or remove (same goes for Figure 4).

Table 9: In general, the consensus for the number of * used to describe statistical significance follows the number of “zeroes” of the P-values. So that P<0.05 or 0.01 --> *, P<0.005 or 0.001 --> **. The authors chose to use *** for P<0.01 and ** for P<0.05, which is a bit confusing, but OK. However, in figure 4 they use *** for P<0.05. Please be consistent with the notation used.

L392-395: Please provide the statistical analysis and values supporting the significance of the difference in the spectral reflectance values between light and dark wing areas. Figure 5 alone is not sufficient.

L425-453: It is unclear where the results described in this section come from. As it stands now, this part is purely descriptive and lacking any statistical support. Please provide the analysis or explain where this information comes from.

Figure 9: The difference between dotted and solid lines in the figure is not clear, as they all are labelled as “light area”? Please provide more information in the figure caption or change the labels so that the figure reads clearly.

L470-472: This is nice! Is this information available also for how fast this temperature is reached (i.e. comparing slopes in the time series analysis)? Same for L487-488.

L 472-474: How can the authors be sure that Sp8-Sp11 store the heat and then transfer it to Sp7? Based on figure 9 Sp7 reaches a higher temperature and has a higher slope (? Table 10) than the other points. But I do not understand how the authors can say which areas absorb the heat and transfer it to other areas? Please explain (same for 488-491).

L489: Replace “storage” by “store”.

L515-544: It is unclear how the results about how the heat is transferred from which point to which point are obtained. For example, in L517-529: are all these data inferred from a table? Please explain your reasoning for readers not familiar with this type of data.

L583: Remove “to”

L588: Replace “branchs” with “branches”.

L609-611: This sentence is not written in correct English, please rephrase.

L699-702: How can the authors say this with certainty? Please explain reasoning in the methods.

---

## Round 0.3 · Minor Revisions

The revisions by the authors have greatly improved the ms according to the previous reviewer and editorial comments. I have received a re-review from one of the previous reviewers that recommends minor revisions. The definitions and use of the terms ‘heat storage’ and ‘heat absorption’ are much clearer now, and the interpretation of the real-time heat experiment is much improved. There are still some details of the heating experiment that are unclear, poorly described or over-worked, and I have provided edits below. Please also consider the suggestions provided by the new review.
Editorial minor revisions:
Line 260-265: the operational definitions read like results rather than your own definitions. Please re-word. I suggest: “Here, we define heat absorption areas as those with fast initial heating rates, but not necessarily high equilibrium temperatures. In contrast, heat storage areas are defined as areas with delayed but rapid heating rates and a high equilibrium temperature.
Line 350, after the comma, delete ‘that’
Line 366, 372: delete ‘significantly’ as there is no statistical test associated with these statements.
Line 369” “The temperatures of the light areas….were lower that those of most of the dark areas ….(Figure 8)”.
Line 371: “…heat appeared to accumulate…” and ref Fig 8 as well.
Line 373-380: Rewrite or delete this section. The FLIR images are compelling visually, but do not provide evidence of specific regions as mentioned here. One cannot remember where all of the cells are in these images. In addition, the fact that the cells are not named the same as the sampling points in the Tables and Figs 8&9 makes it difficult to relate this text to the actual data. The following section of the paper, which covers the data, is far more easy to follow and glean meaning from.
Line 381- delete. One cannot use visual inspection of Fig 7 to conclude anything about heat transfer channels. Rely on the data in the next section to make this point.
Lines 387-413: this is a great improvement on previous versions. However, still some elements are not clear and there is unnecessary information here. First, we do not need the actual values as they are already in the table. Second, do not refer to the ‘dark areas as heat storage’ as they are not defined this way. Third, the presentation for the hind wing can be shorter as it is the same for the forewing. My suggested edit reads:
The temperature at each point on the fore and hind wing surfaces increased as the lighting period increased as the analysis of time-series ExpDec2 of exponential models (Figure 8 & Table 7). On the fore wing, the base of the wing (Sp7) heated at a conspicuously slow rate initially (A1), then heated at a higher rate than other points at later times (A2). This suggests that the base of the wing does not absorb heat but is a heat storage area, as further supported by the high equilibrium temperature parameter (y0). The points in the dark area heated at a high, similar rate initially (A1 of Sp8-Sp12), followed by points not in the border (Sp8−Sp11) continuing to heat at a high rate, while the border point (Sp12) showed a decrease in later heating, suggesting it is poor at heat storage (A2 values).The points in the light area always heated at a similar rate with the points in the dark area, but the equilibrium temperatures were lower than the non-border dark areas (y0s of the points Sp1−Sp6 were significantly lower than the points Sp8−Sp11: t8=2.864, P=0.021), while being similar to those of the border.
In the hind wing, the results were qualitatively the same as for the forewing, with the base of the wing heatingat a conspicuously slow rate initially, followed by a very high late heating rate and high equilibrium value. The non-border dark areas had high heating rates throughout the trial, while the border dark area showed a slow heating rate late in the trial. In contrast to the forewing, we found the equilibrium parameters for the light areas near the base of the wing (y0 of H-Sp1, H-Sp2 and H-Sp3) were similar to those of the non-border dark areas (H-Sp8−H-Sp11; t5=20.683, P=0.525). The possible reason is that these points were close to the base of the wing and were affected by the heat of the base of the wing. The temperatures of other three points in the light area were lower than the non-border dark areas (y0: t5=2.986, P=0.031). In summary, for both wings, the non-border dark areas could temporarily store heat and then transfer it to the base of the wing. The dark area of the border could absorb heat from the light source but it could not store the heat, similar to most light areas on the wing.

Line 422: delete the clause after the semi-colon because it is repetitive.
Line 424-425: delete
Table 2-6: indicate in the footnote whether it is mean +/- SD that is presented.
Table 7: in both places, change “dark areas for heat storage” to “non-border dark areas”

Reviewer 1 ·

Basic reporting

Please see "General comments for the author".

Experimental design

Please see "General comments for the author".

Validity of the findings

Please see "General comments for the author".

Additional comments

The authors made an excellent work in revising the manuscript which reads clearly now.
I only have some minor recommendations aimed at further improving the clarity of the paper. I also recommend removing any mention about the fore wing being longer than the hind wing, as it is not adding any novel piece of information (this is commonly found in butterflies), and cannot be causally linked to thermoregulatory ability in this specific study. In addition, the authors provide only weak evidence about the role of wing veins in heat transfer from heat storage areas to the base of the wing. I encourage the authors to highlight in the discussion that the role of the veins suggested is speculative. Finally, there are several points in the text which are not written in correct English grammar. I pointed out some of these below, and I recommend that the authors perform a thorough grammar check (for example with a grammar check software as “Grammarly”).

Please find detailed comments below.

L 17-23: The abstract reads much clearer now! However, this sentence is extremely long and I recommend to split it into two to enhance readability. For example, you can start a new sentence as “In addition, spectral reflectance…… were also measured”.
In addition “…scales removed or not in light or dark areas” is unclear. Consider revising.

L 23: I suggest to emphasize the results by starting at least the first sentence by “We found that…..”

L 40-41: “….courtship behavior, mating behavior, and oviposition behavior” can be simplified as “courtship, mating, and oviposition behaviorS”

L 55: “….Pieris butterflies use their PREDOMINANTLY WHITE wings….”. To emphasize the role of light wing areas in reflection rather than absorbance.

L 59: “….not only depends on reflection AND ABSORPTION…”

L 83 and elsewhere, including in the reference list at the end: The author is not “Trullas CT”, but “Clusella Trullas S”.

L 99-102: “In THE field”
“show high flight activity on sunny dayS”
I recommend a thorough grammar check with Grammarly.

L 104: “HENCE, according to the color depth….”

L 130: “…or present in light or dark areas, RESPECTIVELY”

L 133: I believe the authors should state something like “once identified the heat storage areas and non-storage areas, we also monitored real-time temperature changes in these areas…….”. Otherwise, it may sound like these areas were decided a priori.

L 147: How many? Please state sample size for males and females.

L 156: “3 days OF AGE”. Same for “4 days” and “7 days”.

L 170: Replace “where this angle” with “which”

L 274: Please add the sample size

L 287: “… evenly distributed POINTS”

L 291: This sentence should be preceded by “we defined heat absorption and heat storage areas based on the following criteria”. The definition of these areas is key to the understanding of the whole study, hence it should be stated as clearly as possible.

L 316: What does “y” represent?

L 329: Just a suggestion. There could be one final sentence stating that the direction of such results is described into detail in the following sections? So that the reader knows what to expect.

Tables 2, 3 and 4: Please specify in the caption the measurement units for temperature, time, and temperature excess.

L 367-370: Again, it is unclear what is the relevance of this finding and why it is included in this paragraph. Forewings being longer than hind wings is extremely common (if not always the case) in butterflies. I suggest removing this result as well as Figure 4.

L 390: Remove “that”

L 433-439 and 446-452: I believe there is a mathematical explanation beyond my understanding for this. However, it is a bit confusing to read about “higher rate” when the values reported in parentheses are increasingly high, but also preceded by a “-“, and therefore “more negative”? Perhaps the authors can clarify this in the methods.

L 534-535: “….requires A CERTAIN optimal wing opening angle THAT CAN VARY DEPENDING ON THE BUTTERFLY SPECIES.”

L 540-541: “In THE wild”
“need heat FOR autonomous fly”

L 545-546: I feel that this should be a fairly common finding in butterflies. Perhaps the authors could cite some other studies showing that forewings are more important that hind wings in thermoregulation?

L 553: Replace “limits” with “impacts”

L 557: Replace “Thus” with “For example, ...”

L 558-561: Again, this is generally the case in butterflies, and not specifically for this species. The authors did not formally test the causal link between variation in wing size and heat absorption and they cannot draw any conclusion about this. Forewings are also the main structure allowing flight, and their length relative to hindwings may also be motivated by that. I suggest the authors remove any statement about the fore wing being larger than the hind wing as irrelevant for the scope of this paper.

L 666-669: I am not convinced the authors have solid evidence that the veins are the mechanism underlying the heat transfer from storage areas to the base of the wing. How can the authors be certain that the heat is transferred through the vein, and not through the wing surface of a certain portion, including ALSO the vein? Perhaps this sentence should begin by “We speculate that…”.
The authors state themselves in the results that “The heat rate of the points in all veins failed to show a certain rule according to the distance from the base of the wing (Figure 9 & 10).” and again that “heat MAY mainly transfer along the veins from the points at the edge of the wing surface to the points at the base of the wing”. Hence, I would say that this sentence is indeed speculation.

---

## Round 0.4 · accepted · Accept

I appreciate your attention to the reviewers' comments on all drafts. The changes you have made according to these comments have improved the manuscript, and it is now suitable for publication.

#